# *Plasmodium* translocon component EXP2 facilitates hepatocyte invasion

João Mello-Vieira [1], Francisco J. Enguita[1], Tania F. de Koning-Ward [2], Vanessa Zuzarte-Luís [1✉] &
Maria M. Mota [1✉]

*Plasmodium* parasites possess a translocon that exports parasite proteins into the infected erythrocyte. Although the translocon components are also expressed during the mosquito and liver stage of infection, their function remains unexplored. Here, using a combination of genetic and chemical assays, we show that the translocon component Exported Protein 2 (EXP2) is critical for invasion of hepatocytes. EXP2 is a pore-forming protein that is secreted from the sporozoite upon contact with the host cell milieu. EXP2-deficient sporozoites are impaired in invasion, which can be rescued by the exogenous administration of recombinant EXP2 and alpha-hemolysin (an *S. aureus* pore-forming protein), as well as by acid sphingo-myelinase. The latter, together with the negative impact of chemical and genetic inhibition of acid sphingomyelinase on invasion, reveals that EXP2 pore-forming activity induces hepa-tocyte membrane repair, which plays a key role in parasite invasion. Overall, our findings establish a novel and critical function for EXP2 that leads to an active participation of the host cell in *Plasmodium* sporozoite invasion, challenging the current view of the establishment of liver stage infection.

[1] Instituto de Medicina Molecular João Lobo Antunes, Faculdade de Medicina, Universidade de Lisboa, 1649-028 Lisboa, Portugal. [2] School of Medicine, Deakin University, Waurn Ponds, Geelong, Australia. ✉email: vluis@medicina.ulisboa.pt; mmota@medicina.ulisboa.pt

A n obligatory step of infection by *Plasmodium* parasites, the causative agents of malaria, is the invasion of host hepatocytes that involves both parasite and host proteins[1]. However, the mechanism by which *Plasmodium* sporozoites enter hepatocytes remains elusive[1]. The major sporozoite surface protein, circumsporozoite protein (CSP), must undergo conformational changes and cleavage[2], likely mediated by *Plasmodium* cysteine proteases. Another surface protein, the thrombospondin-related anonymous protein (TRAP), which is connected to the actin machinery of the parasite, propels the parasite into the hepatocyte[3]. The hepatocyte membrane receptors scavenger receptor class B type I (SR-BI)[4,5] or CD81[6] are also involved, with CD81 being potentially important for the discharge of parasite invasive proteins[7]. The *Plasmodium* 6-cys protein p36[8], has been shown to be important for binding to these host cell receptors[9]. Other host proteins, the ephrin type-A receptor 2[10,11] and the protein kinase C zeta (PKCζ)[12,13] have also been associated with invasion, yet their distinct roles in the invasion process are still unknown. Noteworthy, for years, the prevailing view of *Plasmodium* sporozoite invasion implied that the host cell does not play an active role during parasite entry[14–16], a concept only challenged by the observation that host actin reorganizes around the invading sporozoite[17,18].

Exported Protein 2 (EXP2) is a parasite pore-forming protein located at the parasitophorous vacuole membrane (PVM) of both liver and blood stage parasites[19,20], whose structure has recently been solved[21]. During the blood stage of infection, EXP2 and four other parasite proteins are part of the translocon complex[22], crucial for exporting parasite proteins to the cytosol of the erythrocyte[23,24]. Besides being engaged in protein translocation, recent reports have demonstrated that the EXP2 pore, independent of the other translocon components[25], might also serve as a nutrient channel through which solutes flow between the host cell and the vacuole surrounding the parasite[26,27]. In the liver stage, although EXP2 deficiency leads to a significant decrease in parasite burden[28], its function remains unexplored.

In this work, we use an EXP2 conditional knockout parasite line to study its role during the liver stage of infection. We observe that the lack of EXP2 in the sporozoite stage leads to a decrease in the ability of sporozoites to invade hepatocytes. In fact, EXP2 is detected in sporozoites and is secreted upon stimulation, similarly to other proteins involved in invasion. Our hypothesis is that EXP2 is triggering the host membrane-repair pathway and we observe that the host protein acid sphingomyelinase, critical for this repair process, is also important for sporozoite invasion of hepatocytes. Our results uncover a surprising and critical function of EXP2 in triggering an active response by the hepatocyte, which is key for sporozoite invasion and establishment in the host cell.

## Results

**Conditional knockout of the EXP2 gene in the sporozoite leads to a decrease in the number of infected cells.** To study the function of EXP2 during the liver stage of infection, we used an EXP2 conditional knockout parasite line (EXP2 cKO) based on the flippase recombinase and flippase recognition targets (FLP/FRT) system[29]. In this system, the FLP recombinase expression is controlled either by the TRAP or the Upregulated in Infectious Sporozoite 4 (UIS4) promoter and mediates excision of the FRT sequences flanking the EXP2 3′ UTR, starting at the sporozoite stage of *Plasmodium berghei*. As a control, we used the parental EXP2 FRT parasites lacking the FLP recombinase (WT)[28]. To assess the efficiency of the conditional deletion system, the excision of the EXP2 3′ UTR was quantified by quantitative polymerase chain reaction (qPCR) in sporozoite genomic DNA.

In the parasite line where the FLP expression is controlled by the TRAP promoter excision was very inefficient, with only $26 \pm 12\%$ of EXP2 cKO sporozoites showing an excised EXP2 locus ($p = 0.1429$, Fig. 1a). However, when the UIS4 promoter was used we observed that $45 \pm 4\%$ of EXP2 cKO sporozoites showed an excised EXP2 locus ($p = 0.0286$, Fig. 1a). We selected the UIS4 promoter-induced EXP2 cKO line to be used in subsequent experiments.

The lack of EXP2 protein in sporozoites, the consequence of locus excision, was analyzed in the UIS4 parasite line by microscopy and compared with control WT parasites. We observed that the population of EXP2 cKO sporozoites comprised of sporozoites without the EXP2 protein (EXP2 negative, $42 \pm 7\%$, Fig. 1b, bottom line) and sporozoites with detectable EXP2 protein (EXP2 positive, $58 \pm 5\%$, Fig. 1b middle line), in a proportion similar to that obtained by qPCR. Noteworthy, the EXP2 cKO sporozoite population showed a variety of EXP2 intensity values, ranging from the levels observed in the negative control (unstained) to those observed in WT sporozoites (Supplementary Fig. 1a–b). This suggests that the timing of excision of the EXP2 gene locus varies within the sporozoite population.

To evaluate how the lack of EXP2 impacts liver stage development we infected C57Bl/6 mice with EXP2 cKO sporozoites. When compared with WT parasites, infection with EXP2 cKO sporozoites resulted in a decreased parasite liver load, starting at 6 hours (h) after infection, as detected by qRT-PCR ($57 \pm 7\%$, $p = 0.0056$, Fig. 1c). This decrease was maintained throughout liver stage development ($71 \pm 6\%$ at 24 h, $p = 0.0008$; $62 \pm 10\%$ at 48 h, $p = 0.0037$; Fig. 1c).

Next, we infected HepG2 cells with WT or EXP2 cKO sporozoites and analyzed the number of exoerythrocytic forms (EEFs) and their size by immunofluorescence assay (IFA). Detailed microscopic analysis of HepG2 cells revealed that the number of cells infected with EXP2 cKO sporozoites is reduced at 2 h after infection, when compared with the control parental line ($46 \pm 6\%$, $p = 0.0001$). Once again, this decrease was maintained throughout infection ($47 \pm 5\%$ and $46 \pm 4\%$ at 24 and 48 h after infection, $p = 0.0043$; $p = 0.0024$, respectively, Fig. 1d). Parasite development was not affected (Supplementary Fig. 1c). Notably, the translocon activity during the liver stage is yet to be observed, which might explain why liver stage development is not affected by the lack of EXP2.

Importantly, the early and stable reduction in the number of infected cells at 2 h after infection correlates with the extent of excision of the EXP2 locus at the sporozoite stage (Pearson's $R = 0.880$, Fig. 1e). Conversely, EXP2 expression does not seem to be necessary for the ensuing development, as it was detected in <3% of the remaining EXP2 cKO EEFs, all of which developed normally ($2 \pm 1\%$ at 24 h, $p = 0.0159$; $3 \pm 3\%$ at 48 h, $p = 0.0022$; Fig. 1f and Supplementary Fig. 1d). The increase of the excision rate throughout EEF development can be explained by the fact that the UIS4 promoter is kept active during the liver stage of infection, allowing for more expression of the FLP recombinase and further excision of the EXP2 3′ UTR.

Altogether, these results show that EXP2 is critical for the early establishment of hepatocyte infection by *P. berghei* sporozoites.

**EXP2 cKO parasites are impaired in the invasion of hepatocytes.** Such an early phenotype suggests that EXP2 is important for liver colonization. To successfully establish in the liver, sporozoites use gliding motility to traverse through several host cells prior to invading hepatocytes[30]. Our results show that EXP2 cKO sporozoites glide and traverse similarly to the WT parasites (Fig. 1g–h, respectively). To assess if EXP2 cKO sporozoites were able to infect cells, we performed in-and-out staining at 2 h after

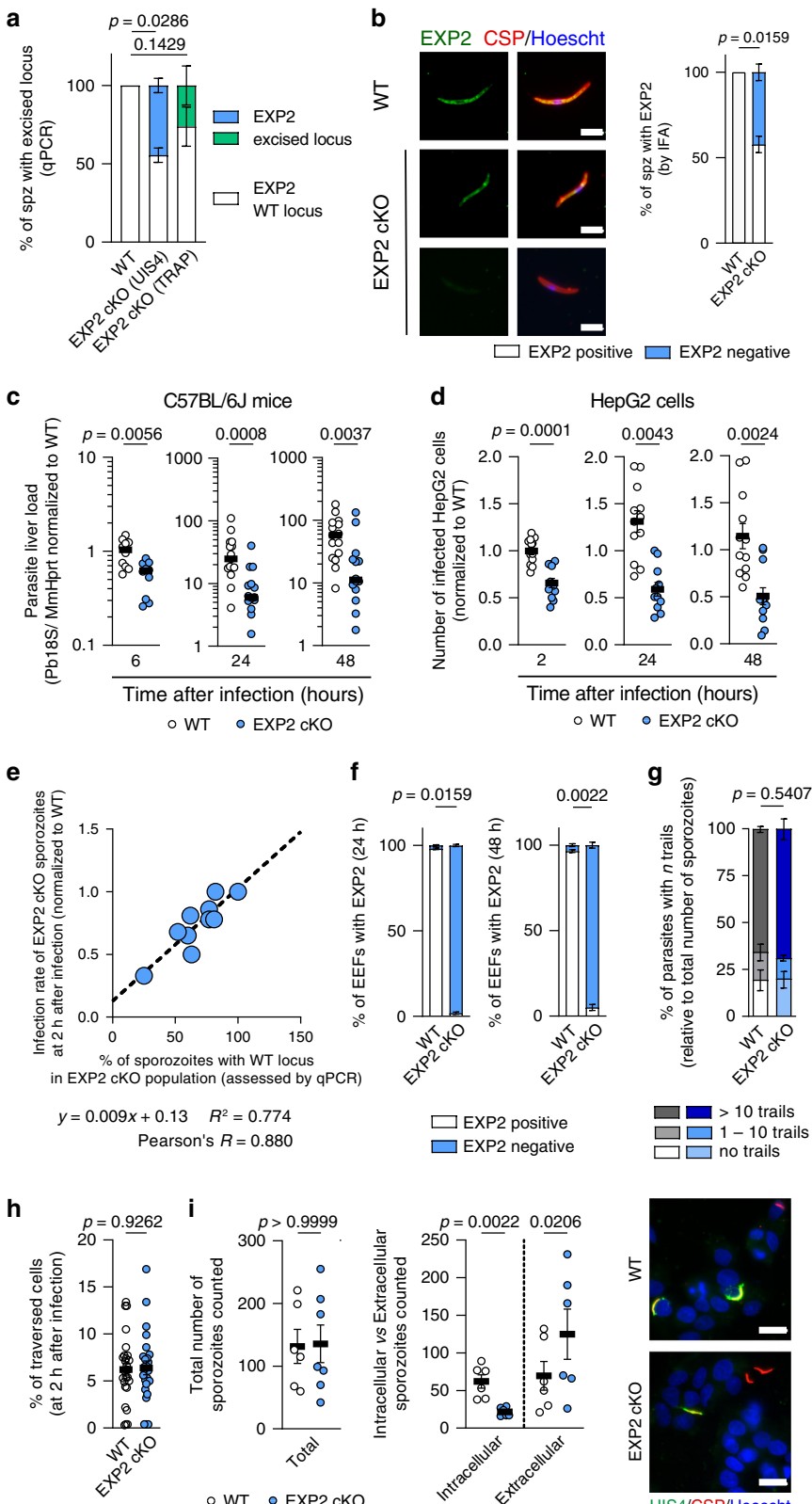

infection. We stained for the sporozoite protein CSP to count the total number of parasites and for the PVM marker UIS4 to assess the number of invaded parasites. We observed that WT and EXP2 cKO sporozoites adhered to cells similarly based on CSP staining (comparable absolute number of sporozoites counted, Fig. 1i). However, EXP2 cKO sporozoites were more predominantly found outside cells (76 ± 21%, $p = 0.0206$) when compared with

WT sporozoites by UIS4 staining (49 ± 13%, $p = 0.0022$), evidencing a defect in hepatocyte invasion (Fig. 1i).

**EXP2 is relocated and discharged by the sporozoite upon contact with the host cell milieu.** Given the invasion phenotype of EXP2 cKO sporozoites, we next sought to characterize EXP2

**Fig. 1 EXP2 deletion hinders invasion of hepatocytes. a** Percentage of sporozoites with WT EXP2 (white) or excised locus (shaded bars) determined by qPCR in WT and EXP2 (UIS4 (blue) or TRAP (green) promoter mediated) cKO sporozoites. ($N = 4$ independent experiments). **b** Micrographs of WT or EXP2 cKO sporozoites, stained with mouse $\alpha Pf$EXP2 (green), $\alpha Pb$CSP (red), Hoechst (blue). Scale bar: 5 µm. Percentage of sporozoites classified as EXP2-positive (white) and EXP2-negative (blue) by immunofluorescence ($N = 7$). **c** Parasite liver load in mice infected with WT (white) or EXP2 cKO (blue) sporozoites by qPCR ($N = 2$ or 3 totaling 10 or 15 mice for 6 h or other timepoints, respectively, normalized to 6 h). **d** Number of cells infected with WT (white) or EXP2 cKO (blue) sporozoites analyzed by immunofluorescence (>30,000 cells analyzed, $N = 4$ or 5 totaling 12 or 15 replicates for 2 h or other timepoints, respectively, normalized to 2 h). **e** Percentage of WT parasites in EXP2 cKO sporozoite population versus number of infected HepG2 cells at 2 h after infection (blue). Proportion of EXP2-positive parasites assessed by qPCR, proportion of invaded parasites quantified by immunofluorescence ($N = 10$ totaling 10 and 30 replicates for excision and invasion rates, respectively). **f** Proportion of EEFs classified as EXP2-positive (white) and EXP2-negative (blue) by immunofluorescence. ($N = 3$). **g** Percentage of WT (gray-shaded) or EXP2 cKO (blue-shaded) sporozoites with CSP trails by immunofluorescence. Sporozoites scored as having 0, between 1 and 10 or >10 trails. ($N = 3$). **h** Percentage of traversed HepG2 cells by WT (white) or EXP2 cKO (blue) sporozoites, assessed by Flow Cytometry. (>10,000 cells analyzed, $N = 8$ totaling 27 and 26 replicates for WT and EXP2 cKO, respectively). **i** Total number of WT (white) or EXP2 cKO (blue) sporozoites (CSP staining), intracellular sporozoites (both UIS4 and CSP) and extracellular sporozoites (only CSP) at 2 h after infection. Micrographs of WT or EXP2 cKO sporozoites at same time point, stained with $\alpha Pb$UIS4 (green), $\alpha Pb$CSP (red), Hoechst (blue). Scale bar: 10 µm. ($N = 3$ totaling 6 and 7 replicates for WT and EXP2 cKO, respectively). Results shown as mean ± SEM, two-tailed Mann–Whitney $U$ test was applied for $p$ values in all panels except for **e**, where correlation and linear regression was performed.

expression and localization in sporozoites. To that end, the localization of EXP2 in freshly dissected *P. berghei* sporozoites was assessed by immuno-EM and we observed that EXP2 is distributed throughout the parasite cell body, frequently observed inside vesicles (Fig. 2a). Micronemes and rhoptries are the most characterized types of vesicles found in *Plasmodium* sporozoites, which contain proteins that participate in the invasion process[31]. We sought to colocalize EXP2 with proteins reported to be present in those organelles, such as TRAP (present in micronemes)[32], RON4 (present in rhoptries)[7] and UIS4 (in still unidentified vesicles)[33,34]. We did not observe colocalization of EXP2 with any of the proteins tested, in freshly dissected sporozoites (Fig. 2b).

Apicomplexan parasites are known to secrete proteins important for the invasion process[35]. To address whether EXP2 is also mobilized during invasion, sporozoites were incubated in medium with fetal calf serum (FCS) at 37 °C, a combination that simulates the host milieu conditions and shown to prime sporozoites for invasion[36–38]. Western Blot (WB) analysis of sporozoites pellet and the supernatant fractions after sporozoite activation (incubation with FCS at 37 °C) showed EXP2, but not the intracellular ER-marker BiP, in the supernatant fraction of activated sporozoites (Fig. 2c). This shows that EXP2 is discharged to the medium upon activation.

The results so far imply that EXP2 relocation and secretion is important for the invasion of hepatocytes. If this hypothesis is correct, the addition of exogenous EXP2 would rescue the invasion impairment of EXP2 cKO sporozoites. Indeed, addition of exogenous recombinant *Plasmodium falciparum* EXP2 (r*Pf*EXP2)[39] to the culture medium restores the invasion capacity of EXP2 cKO sporozoites (Fig. 2d). Interestingly, this rescue only occurs when r*Pf*EXP2 is added to the culture 1 h after sporozoite incubation and not when r*Pf*EXP2 is added to the cells concomitantly with the sporozoites (44 ± 8%, $p < 0.0001$, Fig. 2d) nor if added to sporozoites prior to incubation with cells (43 ± 10%, $p = 0.0022$, Supplementary Fig. 2a).

We sought to understand why EXP2 is needed at 1 h after the sporozoite incubation and not earlier. It has been reported that *Plasmodium* sporozoites begin productive invasion only 1 h after being in contact with cells[40]. We observe that this is also true for spect1 KO sporozoites that are incapable of cell traversal (Supplementary Fig. 2b), suggesting that *Plasmodium* sporozoites require a "waiting period", not related with cell traversal, of ~1 hour, before invading hepatocytes. Interestingly, transcriptomic analysis of activated sporozoites showed EXP2 upregulation starting at 1 h after incubation at 37 °C[41]. Indeed, we observed a peak of EXP2 mRNA expression at 1 h after infection (Fig. 2e). This behavior is specific to EXP2 but not of other invasion-related

genes, such as, Glideosome-associated Protein 45 (GAP45), which showed a tendency to be down-regulated (Fig. 2e), nor that of the liver stage PVM-resident Exported Protein 1 (EXP1), which showed a tendency to be upregulated during invasion (Fig. 2e). In *P. falciparum*, EXP2 was found to be continuously upregulated until 2 h after incubation of sporozoites with cells[41], although traversal has been shown to take longer in *P. falciparum*[42] than in rodent *Plasmodium* parasites[40]. At 2 h after infection, *P. falciparum* is still actively traversing cells[42]. It remains to be tested whether *P. falciparum* EXP2 mRNA decreases after invasion is completed (3–4 h after incubation with cells).

We next sought to examine if the secretion of the EXP2 protein follows the invasion kinetics. Importantly, we observed continuous secretion of EXP2 to the surrounding medium starting at 30 minutes after sporozoite activation and increasing until 2 h after activation (Fig. 2f). These results show that the temporal dynamics of EXP2 expression and secretion accompanies the invasion kinetics observed in *P. berghei* sporozoites.

To complement the previous experiment, we tested if r*Pf*EXP2 has the ability to interact with cellular membranes. For this, we incubated r*Pf*EXP2 with HepG2 cells, collected the cellular medium and we assayed this conditioned medium for the presence of r*Pf*EXP2 by WB. In cells treated with 1 nM of r*Pf*EXP2, we observed a decrease in r*Pf*EXP2 concentration in the supernatant already at 5 min after cells are exposed to the protein (Supplementary Fig. 2c). This decrease is consistent with previous observations for other pore-forming proteins[43], which are readily absorbed by cells. The fact that not all the protein is internalized, and the remaining r*Pf*EXP2 is not absorbed by the cells in the following hour, suggests that cells become non responsive to the action of r*Pf*EXP2.

Remarkably, EXP2 is never detected in sporozoites immediately after invasion (Supplementary Fig. 2d). This behavior has been reported for other proteins important for sporozoite invasion, RON4 and AMA1[7,36], suggesting that EXP2 might be degraded or completely discharged during the invasion process. Altogether, these data strengthen the role of EXP2 during invasion of the hepatocyte.

**EXP2 pore-forming activity induces a membrane-repair pathway, facilitating invasion of sporozoites.** Ad both native and recombinant *Pf*EXP2 protein are known to form pores in membranes[21,39] we wondered if the function of EXP2 during invasion is related to its pore-forming ability. Indeed, EXP2 cKO invasion defect can be rescued by both r*Pf*EXP2 and the recombinant pore-forming protein from *S. aureus*, α-hemolysin (rα-HL) in a similar range of concentrations (Fig. 3a–b). Notably, the pore

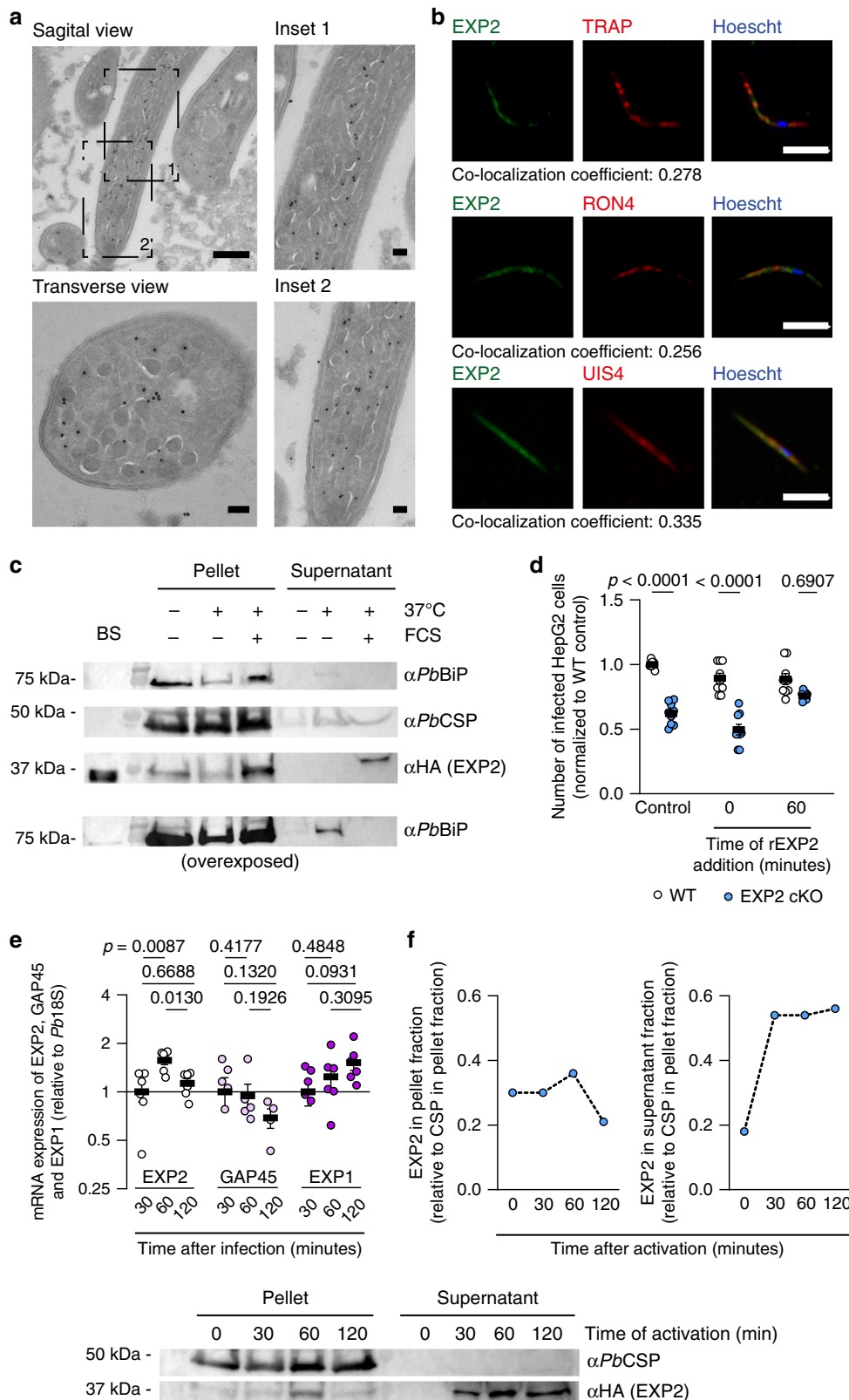

formed by α-HL is similar to that formed by EXP2, both are heptamers[21,44] and create a pore with 1 nm of diameter[21,45]. Importantly, the invasion defect of the EXP2 cKO sporozoites could not be reverted by the *S. pyogenes* pore-forming protein streptolysin O (SLO, Supplementary Fig. 3a) added in a similar concentration. SLO is known to form pores of 25–30 nm in

diameter[46], 10–30 times wider than the pore of EXP2[21] and of α-HL[45].

Host plasma membrane wounding has been previously implicated as a critical step for invasion of *Trypanosoma cruzi* parasites and can also be triggered by the addition of pore-forming toxins to mammalian cells[47]. *T. cruzi*-induced host cell

**Fig. 2 EXP2 is present in the sporozoite and is translocated to the membrane and secreted after activation. a** Immuno-electron microscopy of sporozoites, stained with rabbit αPfEXP2. Scale bar: 500 nm and 100 nm in insets and transverse views. Representative image of three independent experiments. **b** Micrographs of permeabilized sporozoites stained with αPfEXP2 (green), αPbTRAP (microneme protein), αPbRON4 (rhoptry protein), or αPbUIS4 (dense granule protein) (all in red) and the DNA dye Hoechst (blue). For EXP2 co-staining with TRAP and UIS4, mouse αPfEXP2 was used. For co-staining with RON4, rabbit αPfEXP2 was used. Scale bar: 5 μm. (>50 sporozoites analyzed per experiment, N = 3 independent experiments). Shown below each panel is the colocalization coefficient between EXP2 and the corresponding protein. **c** Western blot analysis of secreted EXP2 protein. PbEXP2-HA sporozoites were incubated for 30 min, at 4 °C (−), at 37 °C (+) and in the presence or absence of FCS. Sporozoites were pelleted and both pellet and supernatant were assayed for the presence of PbEXP2-HA, PbCSP (membrane protein of the sporozoite) and PbBiP (ER-resident protein). Lysates of a mixed blood stage infection of PbEXP2-HA were used as control. Representative images of three independent WB. **d** Number of infected cells at 2 h after infection, infected with WT (white) or EXP2 cKO sporozoites (blue), after treatment with 10 nM rEXP2 concomitantly with sporozoite addition to cells or at 1 h after infection (N = 3 independent experiments totaling nine replicates). **e** mRNA expression of invasion-related genes of *P. berghei* at different timepoints during infection, normalized to Pb18S. EXP2 (white), GAP45 (light pink), and EXP1 (dark pink) (N = 2 independent experiments totaling six replicates). **f** Quantification of EXP2 secretion by sporozoites following activation with FCS at 37 °C throughout time. The amount of EXP2 (blue) in pellet or supernatant fractions for each experiment, was normalized to the amount of CSP in the pellet fraction at the respective time point. Protein extraction was performed immediately after sporozoites were removed from incubation (as in Fig. 2c) and secretion was quantified by WB. Representative images of two independent WB. Results in **d** and **e** shown as mean±SEM, two-tailed Mann–Whitney U test was applied for p values.

wounding results in $Ca^{2+}$ influx and consequently exocytosis of lysosomes that deliver acid sphingomyelinase (aSMase) to host plasma membrane, generating ceramide on the outer leaflet of the plasma membrane to facilitate PV formation and internalization of the parasite[48]. We now hypothesize that similarly to *T. cruzi*, *Plasmodium* sporozoites (via EXP2) induce pores in the host cell plasma membrane resulting in $Ca^{2+}$ influx, key for invasion.

The diameter of the EXP2 pore is of 1 nm[21], and it would be wide enough to fit small ions, such as calcium and potassium, the most abundant ions on the outside and inside of mammalian cells, respectively. To assess if rPfEXP2 would allow for the passage of ions we transfected HepG2 cells with the intracellular calcium-sensitive reporter Gcamp6f[49] or the potassium fluorescent sensor GEPII[50]. Treatment of Gcamp6f-transfected cells with rPfEXP2 resulted in an increase in Gcamp6f fluorescence, reflecting an influx of $Ca^{2+}$ (Fig. 3c). In cells transfected with GEPII and treated with rPfEXP2, the fluorescence of the reporter protein decreased, suggesting that potassium is leaking out of the cell (Fig. 3d). It is noteworthy that the concentration of rPfEXP2 in these experiments was 100 nM, ten times higher than that used for the rescue experiments. In the context of invasion, smaller concentrations of rEXP2 might cause local ion changes in the cell that are enough to induce the necessary mechanisms for sporozoite invasion, but not sufficient to induce changes in global ion concentrations.

The next step in the membrane-repair pathway involves the release of host aSMase and its activity in the outer leaflet of the plasma membrane of the hepatocyte. Indeed, when aSMase was inhibited by desipramine[51,52], *P. berghei* sporozoite invasion was impaired ($IC_{50} = 15 \pm 2\,\mu M$) (Fig. 3e). Noteworthy, inhibition of neutral SMase, by GW4869, did not block *P. berghei* sporozoite invasion (Fig. 3e). To further establish host aSMase as a key protein during the invasion process, we proceeded to knockdown its expression using short-hairpin RNA (shRNA). The decrease in aSMase expression by the shRNA ($36 \pm 10\%$, $p = 0.0022$, Supplementary Fig. 3b), resulted in a decrease in *P. berghei* invasion ($56 \pm 17\%$, $p = 0.0022$, Fig. 3f). This effect was counteracted by the addition of recombinant acid SMase from *B. cereus* (Fig. 3f). Remarkably, exogenous addition of recombinant aSMase also rescued the invasion impairment exhibited by EXP2 cKO sporozoites (Fig. 3g).

Altogether, our data show that *Plasmodium* sporozoites secrete EXP2 upon stimulation and that recombinant EXP2 can lead to the formation of pores in the hepatocyte plasma membrane. Both exogenous EXP2 and acid SMase can rescue the invasion impairment of EXP2 cKO sporozoites, suggesting that sporozoites

hijack the host membrane-repair pathway, to facilitate invasion of the host cell.

## Discussion

EXP2 is expressed throughout the *Plasmodium* life cycle[19,20] but has been mostly studied during the blood stages. This protein is found in the dense granules of merozoites[53], which are discharged after the invasion of the erythrocyte is complete. Inside this cell, the parasite relies on EXP2 and the other components of the translocon complex to remodel the erythrocyte with its own proteins[22]. This remodeling is necessary for nutrient acquisition and immune system evasion by the parasite (reviewed in ref. [54]). Beside its association with the translocon complex, EXP2 also exists in regions of the PVM of blood stage parasites without the other translocon components[25], suggesting that EXP2 might have an additional function. Indeed, a recent study suggests that EXP2 contributes to the exchange of small molecules across the PVM, such as amino acids and glucosamine[27]. It seems that these two functions of EXP2 might be influenced by the RON3 protein[55] by a mechanism that is still not understood. Although reports suggest that EXP2 is not involved in erythrocyte invasion[53], we show here that hepatocyte invasion by sporozoites is mediated by EXP2. These data imply not only that *Plasmodium* uses distinct mechanisms to invade hepatocytes and erythrocytes as previously suggested[18], but also that the same parasite molecule plays divergent functions in different stages of infection.

For the invasion of hepatocytes, one hypothesis is that EXP2 is secreted from the sporozoite to the hepatocyte membrane, creating pores that generate ion fluxes, which in turn trigger the release of host aSMase and induces endocytosis, facilitating parasite invasion (Fig. 4). This is in agreement with recent findings showing that *Plasmodium* sporozoites trigger exocytosis of host lysosomes, to facilitate invasion[56]. Notably, other pathogens including another eukaryotic parasite, *T. cruzi*[48], but also the bacteria *Listeria monocytogenes*[57] and *Neisseria gonorrhoeae*[58] as well as human adenoviruses[59] have also been reported to invade cells by hijacking host cell plasma membrane-repair pathways resulting in ceramide generation. These ceramide membrane domains have been shown to bind and activate PKCζ[60], a kinase that activates endocytosis[61], and that has been previously identified as a positive modulator of *P. berghei* invasion[12] and establishment[13]. Moreover, ceramide generation increases the rigidity of the plasma membrane, leading to the aggregation of host cell receptors, namely CD81[62]. CD81 has been shown to be essential for invasion of *P. falciparum* and *P. yoelii*[6], possibly by inducing rhoptry discharge and moving junction formation[7].

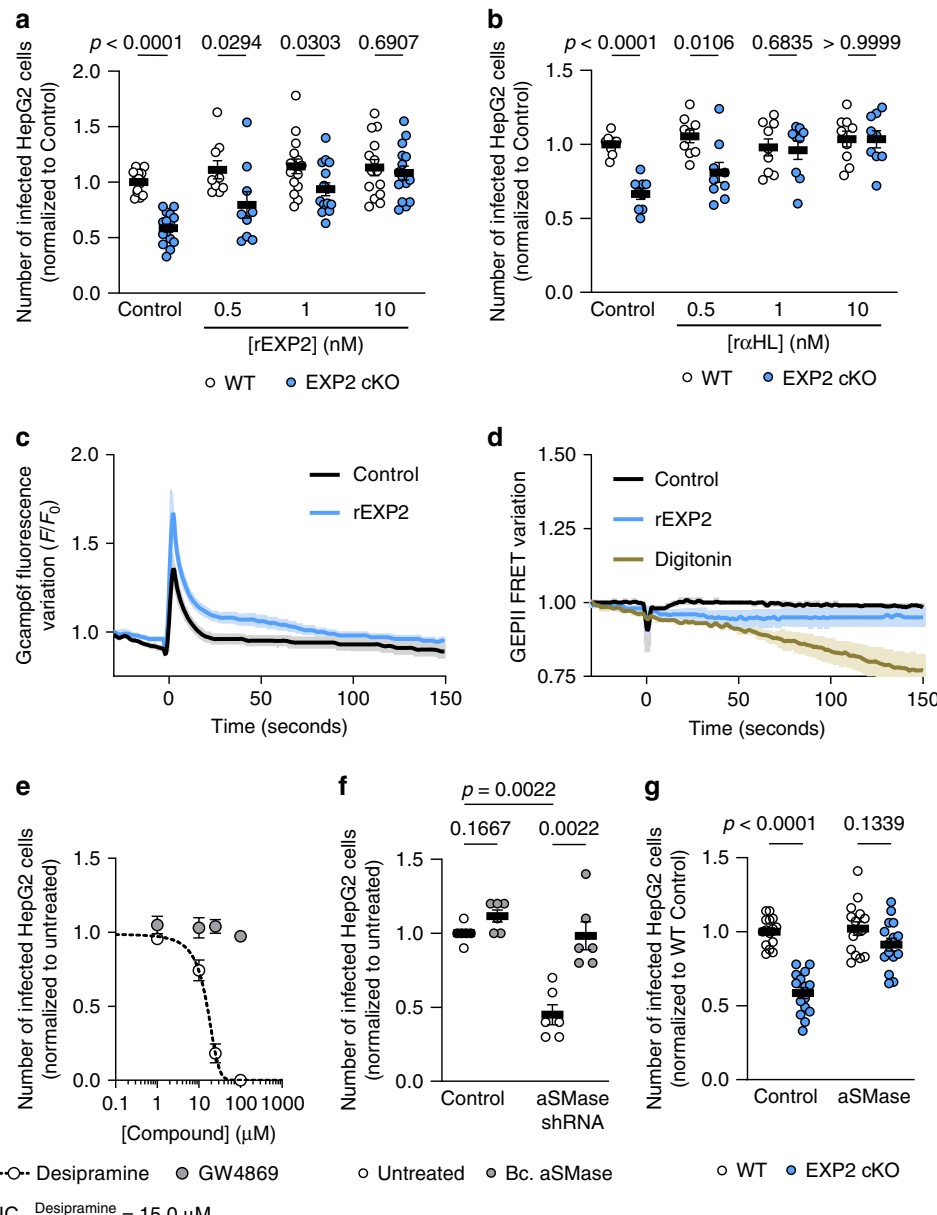

**Fig. 3 Hepatocyte membrane repair is important for invasion of *Plasmodium* sporozoites. a** Number of infected cells at 2 h after infection, infected with WT (white) or EXP2 cKO sporozoites (blue), after addition of r*Pf*EXP2 at 1 h after infection ($N = 5$ independent experiments totaling 15 replicates for all concentrations except for 0.5 nM, where $N = 3$ independent experiments totaling nine replicates were performed). **b** Number of infected cells at 2 h after infection, infected with WT (white) or EXP2 cKO sporozoites (blue), after addition of rα-HL at 1 h after infection ($N = 3$ independent experiments totaling nine replicates). **c** Calcium uptake by cells measured using Gcamp6f fluorescent protein, through time after addition (at time 0 s) of 100 nM r*Pf*EXP2 (blue line) or vehicle control (black line) ($N = 3$ independent experiments, >75 total cells imaged per condition). **d** Potassium efflux by cells measured using GEPII fluorescent protein, through time after addition (at time 0 s) of 100 nM r*Pf*EXP2 (blue line), 50 μM of Digitonin (brown line, permeabilizes cellular membranes) or vehicle control (black line) ($N = 2$ independent experiments, >50 cells imaged per condition). **e** Number of infected cells at 2 h after infection by GFP-expressing sporozoites in the presence of desipramine (acid SMase inhibitor, white) or GW4869 (neutral SMase inhibitor, gray) ($N = 3$ independent experiments totaling nine replicates). **f** Number of infected cells at 2 h after infection by GFP-expressing sporozoites after knockdown of aSMase by shRNA, in the absence (white) or presence (gray) of 10 μU/mL of recombinant acid SMase from *Bacillus cereus* (gray). ($N = 3$ independent experiments totaling nine replicates). **g** Number of infected cells at 2 h after infection, infected with WT (white) or EXP2 cKO sporozoites (blue), after treatment with 10 μU/mL of recombinant aSMase added at 1 h after infection. ($N = 5$ independent experiments totaling 15 replicates). Results are shown as mean±SEM, two-tailed Mann–Whitney $U$ test was applied for $p$ values in all panels except for **c**, **d**. **c**, **d** Lines represent the average of the mean fluorescence intensity and shaded areas the 95% confidence intervals for each time point. Dose-inhibition curve in **e** was generated and $IC_{50}$ value was estimated from the fitted curve.

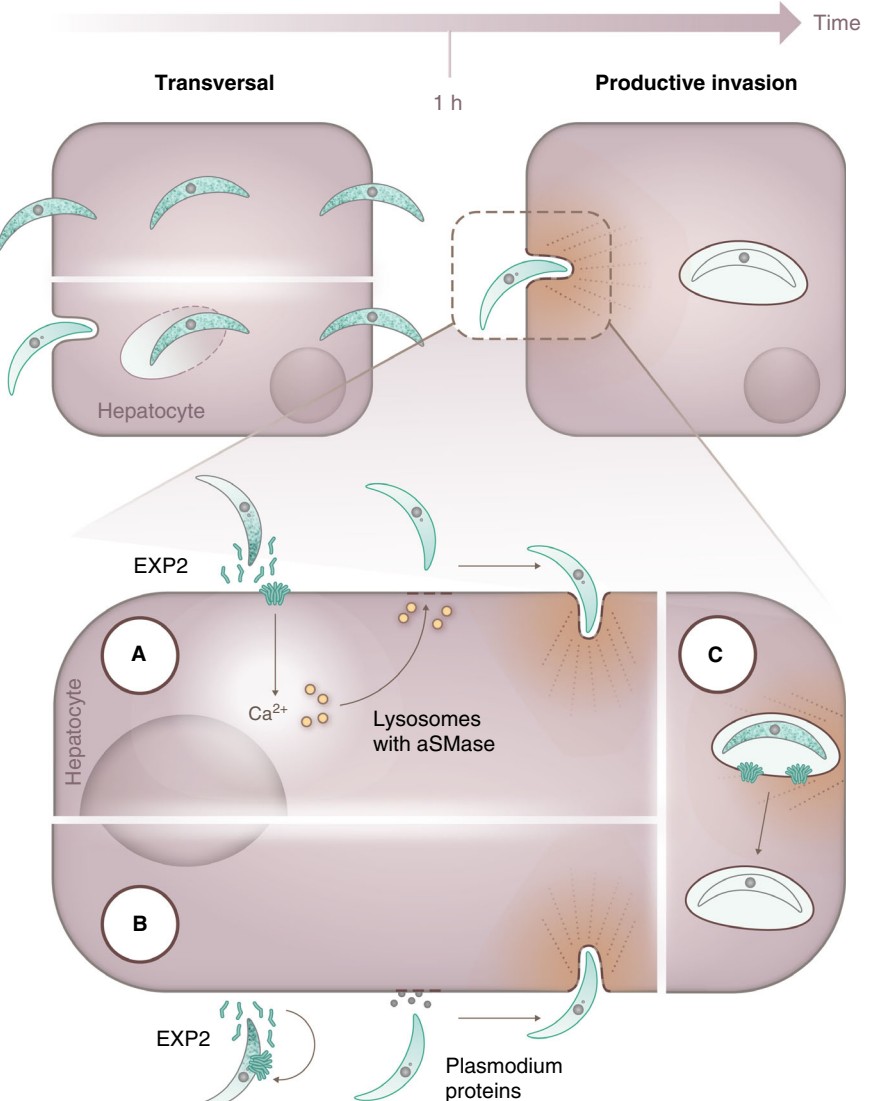

**Fig. 4 Model for the action of *Plasmodium berghei* EXP2.** *Plasmodium berghei* sporozoites, after reaching the liver, traverse hepatocytes before invading the final one inside which they will replicate. Once they encounter the definitive hepatocyte, and stimulated for ~1 hour by the host cell milieu conditions (increase in temperature and presence of mammalian serum components), sporozoites secrete the pore-forming protein EXP2. **a** EXP2 might create pores at the membrane of hepatocytes, allowing the influx of calcium, leading to the activity of host aSMase that facilitates the invasion of the hepatocyte by the sporozoite. **b** Alternatively, EXP2 might bind to the sporozoite membrane, inducing a calcium influx in the sporozoite, which triggers the release of rhoptry and other proteins, allowing for the sporozoite to invade the hepatocyte. **c** Another possibility is that EXP2 is discharged by the sporozoite during vacuole formation, similarly to what happens during the blood stage, being critical for the survival of the sporozoite during the initial stages of hepatocyte infection.

Importantly, increasing the rigidity of the membrane facilitates the invasion of hepatocytes by sporozoites by clustering CD81 at the membrane of the hepatocyte[63]. As such, the hijack of the membrane-repair pathway by *Plasmodium* sporozoites might lead to both aggregation of host cell receptors essential for invasion as well as a trigger for endocytosis via PKCζ.

We cannot exclude that EXP2 secretion might also create pores at the surface of the sporozoite, leading to the increase of intracellular calcium in the sporozoite, triggering the release of more invasion proteins. Nonetheless, this EXP2-induced calcium increase would need to be timed with the expression of other invasion proteins, as the addition of rPfEXP2 to unactivated sporozoites does not rescue the invasion capacity of EXP2 cKO (Fig. 4).

On the other hand, our sporozoite secretion assay does not allow us to distinguish between pre-invasion and post-invasion release. As such, it is also possible that EXP2 might be incorporated in the nascent PVM, not being required for the invasion process itself but for vacuole integrity and parasite maturation during the few minutes/hours after the parasite has sealed the PVM, similarly to what happens during the blood stage of *Plasmodium* infection. Given that we rescue the invasion defect with another pore-forming protein from a different pathogen, this suggests that the role for EXP2 in this maturation process can be attributed to its pore-forming ability. Noteworthy, the action of EXP2 in this scenario would be transient, as we do not detect EXP2 in the PVM following invasion of the hepatocyte. Moreover, parasites lacking EXP2 would still need to escape this immature vacuole and leave the hepatocyte, as EXP2 cKO sporozoites are found preferably outside cells. In essence, in this scenario, EXP2 could be a parasite factor that transforms a transient vacuole into a definitive one, a concept that has been put forward recently[40].

Interestingly, our data also show that the invasion process of *Plasmodium* sporozoites requires a waiting period before the parasite commits to invasion of the hepatocyte. This has been observed before[40] and possibly correlates with the synthesis and secretion of invasion proteins, such as P36[8] and likely EXP2. In fact, sporozoites have been shown to possess a mechanism of translational repression, which prevents the production of liver stage proteins when the sporozoite is in the mosquito salivary glands[64]. Importantly, EXP2 was found to be upregulated in the absence of the mRNA binding protein Puf2, a parasite protein responsible for this process[64]. It is possible that this waiting period correlates with the release of a regulator of sporozoite invasion from translation repression, which initiates transcription of the EXP2 gene. In fact, addition of r*Pf*EXP2 to sporozoites before this waiting period is complete does not rescue the defect of EXP2 cKO. This suggests that this waiting period is sporozoite-intrinsic and that premature presence of EXP2 does not accelerate it. Sporozoite invasion likely requires the timely expression and secretion of a number of parasite proteins, including EXP2. Particularly, we think that this waiting period could occur in the dermis, where sporozoites are deposited during the mosquito bite, to avoid infection of skin cells, where parasite development is less productive[65]. Indeed, it has been shown that sporozoites can remain in the dermis for up to 1 hour after being deposited by the mosquito bite[66], a time frame consistent with that observed by us and others[40].

Finally, we also observed that the lack of EXP2 does not seem to affect EEF development. This observation contrasts with the essential role of EXP2 in the blood stages. Notably, the translocon activity during the liver stage is yet to be observed. In fact, a study showed that a green fluorescent reporter protein, which is exported by the translocon during the blood stage of infection, is retained in the parasite during the liver stage[28]. Moreover HSP101, the ATPase that powers the translocon and unfolds cargo[67], is not expressed during the liver stage[19,28]. This suggests that protein export during the liver stage might function independently of EXP2 and the translocon.

Overall, we suggest that by forcing the host cell to respond to the membrane damage caused by EXP2, the hepatocyte has an active role in *Plasmodium* sporozoite invasion. Importantly, althouhgh EXP2 has divergent functions throughout *Plasmodium* life cycle, our findings point to the convergent evolution of different intracellular pathogens, which have developed similar strategies to take advantage of cellular responses to membrane damage, hitchhiking their way into the host cell.

## Methods

**Mice**. Male C57BL/6 J and BALB/c wild-type mice, aged 6–8 weeks, were purchased from Charles River Laboratories (Saint-Germain-sur-l'Arbresle, France). Mice were housed in the facilities of the iMM João Lobo Antunes (iMM JLA), in specific pathogen-free environment and given water and food ad libitum. Mice are kept in 22–24 °C and 45–65% humidity in a 14h-light/10h-dark cycle. All in vivo protocols were approved by the ORBEA committee of the iMM JLA and were performed according to national and European regulations.

**Sporozoite production**. The following parasite lines were used in this study: GFP-expressing *P. berghei* ANKA (clone 259cl2); *P. berghei* NK65 EXP2 FRT (WT), *P. berghei* NK65 EXP2 FRT UIS4 FLP Recombinase (EXP2 cKO), *P. berghei* NK65 EXP2 FRT TRAP FLP Recombinase and *P. berghei* NK65 EXP2-HA. The first was a kind gift from Chris Janse from Leiden University Medical Center from Leiden, The Netherlands, whereas the other lines were generated in the laboratory of Tania de Koning-Ward from Deakin University, Australia, and are described in a previous study that also outlines the protocol to achieve excision of the EXP2 3′ UTR[28]. Parasites were stored in frozen blood vials, containing $10^7$ blood stage parasites, and kept at −80 °C in our laboratory. To achieve sporozoites, $10^7$ infected red blood cells were injected intra-peritoneally into a BALB/c wild-type mouse. After 5 days of infection, exflagellation of the male gametes in the blood of infected mice was observed under a light microscope. If more than five events per field of view were observed, the infected mouse was used to feed naive *Anopheles*

*stephensi* mosquitos, bred in the insectary of the iMM JLA, for 30 min. For the *P. berghei* NK65 EXP2 FRT TRAP FLP Recombinase parasite line, infected mosquitos were placed at 25 °C 18 days after the mosquito bite, to enhance the activity of the recombinase[29]. In total, 22–35 days after the mosquito blood meal, salivary glands containing *P. berghei* sporozoites, were dissected from infected female *Anopheles stephensi* mosquitoes into simple Dulbecco's Modified Eagles Medium (DMEM, Gibco, Thermo Fisher Scientific, Waltham, MA, USA) and collected into an Eppendorf tube (Eppendorf, Hamburg, Germany). Salivary glands were smashed with a plastic pestle and filtered through a 40-μm Falcon cell-strainer (Thermo Fisher Scientific) to release sporozoites. Sporozoites were counted using a hemo-cytometer (Marienfeld Superior, Lauda-Königshofen, Germany).

**Hepatoma cells**. HepG2 cells (obtained from ATCC, Manassas, VA, USA) were cultured at 37 °C, with a 5% $CO_2$ atmosphere, in DMEM (Gibco), supplemented with 10% FCS (Gibco), 1% glutamine (Gibco) and 1% penicillin/streptomycin (Gibco). Cells were seeded onto no. 1 grade 12 mm diameter glass coverslips (VWR, Radnor, PA, USA) in 24-well plates (Thermo Fisher Scientific) or onto black glass-bottom 96-well plates (Greiner, Kremsmünster, Austria). After plating, cells were incubated at 37 °C, with a 5% $CO_2$ atmosphere.

**Recombinant EXP2 protein production and purification**. Plasmid for recombinant EXP2 protein (rEXP2) production was a kind gift from Professor Masafumi Yohda from Tokyo University of Agriculture and Technology from Tokyo, Japan[39]. In brief, the plasmid pGEX-3X-HRV 3C-EXP2 containing a truncated version of the EXP2 protein fused to Glutathione S-transferase (GST) and an HRV 3C protease cleavage site to excise the affinity tag, was transformed into *E. coli* B21, a gift from the laboratory of Gonçalo Bernardes at iMM JLA. Transformed bacteria were grown in LB broth supplemented with 100 μg/mL ampicillin at 37 °C and when the optical density ($OD_{600}$) of the culture reached 0.5, the protein expression was induced with isopropyl b-D-1thiogalactopyranoside to a final concentration of 0.5 mM, and further incubated for 24 h at 18 °C, after which they were pelleted by centrifugation.

The pelleted bacteria were resuspended in the PBS containing 137 mM NaCl (Sigma); 2.7 mM KCl (Sigma); 1 mM EDTA (Sigma), and 0.15 mM phenylmethylsulfonyl fluoride (Sigma) and disrupted by sonication. Cell debris was eliminated by centrifugation at $14,000 \times g$, the supernatant collected and filtered through a membrane with 0.45 μm pore size and incubated with 0.05% N-dodecyl-b-D-maltopyranoside (DDM, Sigma) at 4 °C for 1 h prior to loading onto a 1 mL GST Hitrap FF column (GE Healthcare, Chicago, IL, USA). The column was then washed with a 10-fold volume of washing buffer (PBS with 0.02% DDM). To release rEXP2, HRV 3 C protease (Takara Bio, Shiga, Japan) in 50 mM Tris-HCl, 200 mM NaCl and 0.02% DDM was applied to the column. After a 16 h incubation at 4 °C, rEXP2 was eluted from the column with PBS buffer. All the purification steps were performed in an AKTA Explorer chromatographic system (GE Healthcare, Chicago, IL, USA). rEXP2 concentration was estimated using Pierce BCA Protein Assay Kit (Thermo Fisher Scientific).

**Calcium imaging using Gcamp6f**. HepG2 cells were transfected with Gcamp6f calcium reporter fluorescent plasmid (Addgene Plasmid #40755, Watertown, MA, USA, a gift from Mafalda Pimentel at iMM JLA) using FuGene 6 HD (Promega, Madison, WI, USA) and OptiMEM (Gibco). In total, 48 h after transfection, cells were imaged in Zeiss Cell Observer widefield fluorescent microscope for 3 min, using ZEN 2 software (Blue version). After an initial 30 s of imaging, the appropriate concentration of rEXP2 or vehicle control was added to the culture medium and cells were imaged for another 150 s. Fluorescence signal through time was normalized to the fluorescence at time −30s ($F_i/F_{−30}$).

**Potassium imaging using GEPII sensor**. HepG2 cells were transfected with GEPII potassium reporter fluorescent plasmid (NGFI, Graz, Austria) using FuGene 6 HD (Promega, Madison, WI, USA) and OptiMEM (Gibco). In total, 48 h after transfection, cells were imaged in Zeiss LSM 710 confocal fluorescent microscope for 3 min, using ZEN 2 software (Blue version). After an initial 30 s of imaging, the appropriate concentration of rEXP2, digitonin or vehicle control was added to the culture medium and cells were imaged for another 150 s. Cells were excited using the 405 nm laser and fluorescence was collected using the following bandwidths: 420–480 nm for CFP channel and 500–700 nm for FRET channel. To calculate the FRET ratio, the intensity of the FRET channel was divided by the intensity of the CFP channel through time. These values were further normalized to the fluorescence at time −30s ($F_i/F_{−30}$).

**Chemicals and proteins**. Desipramine, recombinant α-hemolysin (α-HL) from *Staphylococcus aureus*, recombinant streptolysin O (SLO), Digitonin and recombinant acid Sphingomyelinase (aSMase) from *Bacillus cereus* was purchased from Sigma (Kawasaki, Japan).

**Liver infection and parasite burden determination**. For sporozoite infections, mice were injected with $2 \times 10^4$ sporozoites, in 200 μL of DMEM via intravenous injection. At different timepoints after sporozoite infection (6, 24, 48 h after

infection), livers of infected mice were collected into 3 mL of denaturing solution (4 M guanidium thiocyanite (Sigma), 25 mM sodium citrate (Sigma), 0.5% sarcosyl (Sigma) in MilliQ water treated with DEPC (Sigma). Livers were then mechanically homogenized using 1 mm diameter silica beads (BioSpec Products, Bartlesville, OK, USA) in MiniBeadBeater homogenizer (BioSpec Products) for 2 min. After mechanical disruption of the tissue, 100 μL of homogenate was used to extract RNA, using NZY Total RNA Isolation Kit (NZYTech, Lisboa, Portugal), as per manufacturer's instructions. In all, 1 μg of extracted RNA was converted into cDNA using NZY First-Strand cDNA Synthesis Kit (NZYTech), as per manu-facturer's instructions. cDNA was then used for quantitative Polymerase Chain Reaction (qPCR), by measuring the abundance of *Pb*18s RNA, compared with *MmHprt* RNA using either ViiA 7 (384-well plates, using QuantStudio, v1.3 software) or 7500Fast (96-well plates, using 7500Fast, v2.3 software) Real-Time PCR Systems (Thermo Fisher Scientific) using iTaq Universal SYBR Green Supermix (Bio-Rad Laboratories, Hercules, CA, USA). Analysis of results was performed using the ΔΔC$_T$ method after export of C$_T$ values from the collection softwares:

$$\Delta C_t = C_t^{Gene\ of\ interest} - C_t^{Housekeeping} \qquad (1)$$

$$\Delta \Delta C_t = \Delta C_t^{Experimental} - \Delta C_t^{Control} \qquad (2)$$

$$Relative\ Gene\ expression = 2^{[\Delta\Delta C_t]} \qquad (3)$$

Primers used for liver load determination:
*Pb*18s – forward primer: AAGCATTAAATAAAGCGAATACATCCTTAC
*Pb*18s – reverse primer: GGAGATTGGTTTTGACGTTTATGTG
*MmHprt*– forward primer: TTTGCTGACCTGCTGGATTAC
*MmHprt*– reverse primer: CAAGACATTCTTTCCAGTTAAAGTTG

**Infection of cells**. To infect cells, the required number of sporozoites were sus-pended in complete DMEM and incubated with cells and centrifuged for 5 min at 1 600 × *g*. Cells were incubated for 2 hours at 37 °C, with a 5% CO$_2$ atmosphere, after which the medium was replaced with complete DMEM supplemented with 0.3% Fungizone (Gibco). For some experiments, compounds or rEXP2 protein was added during infection. These materials were either added to the cells con-comitantly with sporozoites or added to the cells 1 hour after the sporozoites, as described in the figure legends.

**Expression of parasite genes during invasion**. After sporozoites and cells were incubated for the appropriate amount of time, mRNA from samples was extracted using NZY Total RNA Isolation Kit (NZYTech), as per manufacturer's instruc-tions. In all, 1 μg of extracted RNA was converted into cDNA using NZY First-Strand cDNA Synthesis Kit (NZYTech), as per manufacturer's instructions. cDNA was then used for qPCR, by measuring the abundance of *Pb*EXP2, *Pb*EXP1 and *Pb*GAP45 was compared with *Pb*18s RNA using either ViiA 7 (384-well plates, using QuantStudio software, v1.3) or 7500Fast (96-well plates, using 7500Fast software, v2.3) Real-Time PCR Systems (Thermo Fisher Scientific) using iTaq Universal SYBR Green Supermix (Bio-Rad Laboratories, Hercules, CA, USA). Analysis of results was performed using the ΔΔC$_T$ method as described above.
Primers used for determination of expression of *Plasmodium* invasion genes:
*Pb*18s – forward primer: AAGCATTAAATAAAGCGAATACATCCTTAC
*Pb*18s – reverse primer: GGAGATTGGTTTTGACGTTTATGTG
*Pb*EXP2 – forward primer: ACGATCCAGGTTTGATTG
*Pb*EXP2 – reverse primer: TGGTAATAGTGGGACATTC
*Pb*GAP45 – forward primer: GTGGAGTAGTCTTTAAGG
*Pb*GAP45 – reverse primer: GTGGAGTAGTCTTTAAGG
*Pb*EXP1 – forward primer: AGGGAAGACATCCATTCCAAATTGG
*Pb*EXP1 – reverse primer: TGAAGATTTGGCATGTTAAGTGGTG

**IFA of hepatoma cells**. To process cells and parasites for IFA, experiments were performed either on black glass-bottom 96-well plates or on no. 1 grade 12 mm diameter glass coverslips. Cells and parasites were fixed at the appropriate time point in 4% paraformaldehyde (Santa Cruz Biotechnology, Dallas, TX, USA) for 10–20 min at room temperature. Cell and parasite fixed material were then blocked and permeabilized with 5% bovine serum albumin (BSA, NZYTech), 0.2% saponin (Sigma) in PBS for at least 30 min at room temperature. Cells and parasites were stained with the appropriate primary antibodies (diluted in 5% BSA, 0.2% saponin in PBS) at room temperature in a humid chamber for 2 h at room temperature in a humid chamber. Samples were washed three times with PBS for 10 min. After washing, samples were stained with appropriate fluorescent secondary antibodies (Thermo Fisher Scientific) (diluted 1:300 in 5% BSA 0.2% saponin in PBS) and with 1 mg/mL Hoechst 33342 (Thermo Fisher Scientific) for 1 hr at room temperature in a humid chamber. Stained samples were washed three times with PBS for 10 min. After the final wash, cov-erslips were mounted on glass slides using a drop of Fluoromount-G (Thermo Fisher Scientific). If black glass-bottom 96-well plates were being used, 50 μL of Fluoromount G was placed on the stained well.
For in and out experiments, fixed samples were incubated with α*Pb*CSP antibody before permeabilization was performed, to stain parasites that would be outside cells. After incubation with α*Pb*CSP antibody, samples were washed with

PBS and were permeabilized and blocked using 5% BSA, 0.2% saponin in PBS. After permeabilization, samples were incubated with α*Pb*UIS4 antibody and the staining progressed as explained above. More than 40 sporozoites were imaged per replicate per parasite line.
For time-course analysis, samples were imaged using either Leica DM5000B (Leica, Wetzlar, Germany), Zeiss Axiovert 200 M or Zeiss Cell Observer (Zeiss, Oberkochen, Germany), which are all widefield fluorescence microscopes, using a ×20 dry objective. Data were collected using Leica DFC Twain (v7.7.1), Metamorph (v7.7.9.0), and ZEN 2 (Blue version), respectively.
The primary antibodies used were goat α*Pb*UIS4 (1:1000 dilution, Sicgen, Cantanhede, Portugal); mouse α*Pb*CSP (1:1000 dilution, clone 3D11, Malaria Research and Reference Reagent Resource Center, MR4, BEI Resources, ATCC); mouse α*Pf*EXP2[68] (1:500 dilution, clone 7.7, The European Malaria Reagent Repository, Edinburgh, Scotland, UK); rabbit α*Pf*EXP2[53] (1:500 dilution, a kind gift from the laboratory of Brendan Crabb).

**Sporozoite staining**. To stain for sporozoite proteins, $5 \times 10^4$ freshly dissected sporozoites were incubated in DMEM medium or DMEM containing 10% FCS at 37 °C on glass coverslips. After 30 min of incubation, sporozoites were fixed in 4% paraformaldehyde for 10–20 min at room temperature. Fixed sporozoites were permeabilized using ice-cold methanol for 5 min at −20 °C. After washing, spor-ozoites were blocked and stained as described above. Coverslips were imaged using Zeiss LSM 710 confocal laser point-scanning fluorescence microscope (using ZEN 2 software, Blue edition), using ×63 or ×100 oil objective. The primary antibodies used were goat α*Pb*UIS4 (1:500 dilution, Sicgen, Cantanhede, Portugal); mouse α*Pb*CSP (1:1000 dilution, clone 3D11, MR4); mouse α*Pf*EXP2 (1:300 dilution, clone 7.7, The European Malaria Reagent Repository, Edinburgh, Scotland, UK); rabbit α*Pf*EXP2 (1:300 dilution, a kind gift from the laboratory of Brendan Crabb[53]) and rabbit α*Pb*TRAP (1:1000 dilution, a kind gift from the laboratory of Joana Tavares[69]) and mouse α*Pb*RON4 (1:300 dilution, a kind gift from the laboratory of Maryse Lebrun[70]).
For colocalization experiments, the plug-in Coloc 2 from FIJI image analysis software was used.

**Quantification of EXP2-positive and EXP2-negative sporozoites in EXP2 cKO sporozoite population**. To distinguish which sporozoites would be EXP2-sufficient and EXP2-deficient, immunofluorescence analysis was performed in WT or EXP2 cKO freshly dissected sporozoites stained with αEXP2 antibody. To assess the background fluorescence intensity of the staining, we also imaged sporozoites that were not stained with αEXP2 antibody. This allowed us to create a cutoff value for the intensity of EXP2 staining, below which EXP2 cKO sporozoites would be considered as EXP2-deficient (see Supplementary Fig. 1a–b for an example of how sporozoites were analyzed). More than 50 sporozoites or EEFs were imaged per replicate per parasite line.

**Gliding assay**. For gliding assays, $5 \times 10^4$ sporozoites were incubated in complete DMEM medium at 37 °C on glass coverslips. After 30 min of incubation, spor-ozoites were fixed in 4% paraformaldehyde for 10–20 min at room temperature. Fixed sporozoites were blocked as described above and only stained with αCSP antibody (1:1000 dilution, clone 3D11) and an anti-mouse secondary antibody (1:300 dilution) and Hoechst. Coverslips were imaged using a Leica DM5000B widefield fluorescence microscope and ×40 dry objective, using Leica DFC Twain (v7.7.1) software. Since CSP is shed in a circular fashion during sporozoite gliding, sporozoite motility can be quantified by counting the circles of CSP (termed trails) produced by each individual sporozoite. Sporozoites were divided into three groups (no trails; 1–10 trails; >10 trails) based on the number of trails that it displayed (see Supplementary Fig. 1e for an example of how sporozoites were separated between these groups). More than 75 sporozoites were imaged per replicate per parasite line.

**Flow cytometry analysis of cell traversal**. To quantify the level of traversal, HepG2 cells and sporozoites were incubated in the presence of 0.5 mg/mL of 10 kDa Dextran-Rhodamine (Thermo Fisher Scientific) in complete DMEM at 37 °C. The dextran molecule is passively taken up by cells that have been traversed and is detected because of the Rhodamine dye, which has Excitation/Emission maxima at 570/590 nm. Data collection was performed using BD Accuri C6 cytometer (Franklin Lakes, New Jersey, USA) and Accuri C6 software (v1.0.264.21) software was used. Data analysis was performed using FlowJo X software (FlowJo LLC, Ashland, OR, USA) (see Supplementary Fig. 1f for the gating strategy used).

**Sporozoite genomic DNA purification**. After dissection from salivary glands, sporozoites were pelleted by centrifugation in a table top centrifuge (Eppendorf) at maximum speed for 15 min. Genomic DNA was purified using NZY Blood gDNA Isolation kit (NZYTech). Gene abundance in sporozoite genomic DNA was ana-lyzed using either ViiA 7 (384-well plates, using QuantStudio v1.3 software) or 7500Fast (96-well plates using 7500Fast v2.3 software) Real-Time PCR Systems (Thermo Fisher Scientific) using iTaq Universal SYBR Green Supermix (Bio-Rad). Analysis of results was performed using the ΔΔC$_T$ method described above.
Primers used for assessing EXP2 recombination/excision:
*Pbdhfr* – forward primer: GTTGGTTCGCTAAACTGCATC

*Pbdhfr* – reverse primer: CTGTTTACCTTCTACTGAAGAGG
*Pbhsp70* – forward primer: TGCAGCAGATAATCAAACTC
*Pbhsp70* – reverse primer: ACTTCAATTTGTGGAACACC

**Blood stage parasite protein isolation**. For preparation of samples from blood stages, BALB/c mice were infected with *P. berghei* NK65 EXP2-HA (*Pb*EXP2-HA) and parasites multiplied until reaching a parasitemia of ~10%. Mice were sacrificed using isoflurane (Abbott Laboratories, Lake Bluff, IL, USA) overdose and blood was removed by heart puncture. Red blood cells were suspended in PBS and pelleted by centrifugation at $450 \times g$ for 5 min without brake. Pelleted red blood cells were lysed using 0.05% Saponin containing cOmplete Protease Inhibitor Cocktail (Sigma) in PBS for 5 min of ice. Parasites were pelleted at $2500 \times g$ for 5 min and were lysed in 500 μL of radioimmunoprecipitation assay (RIPA) buffer buffer with 1% sodium dodecyl sulphate (SDS) for 1 h on ice.

**Sporozoite secretion assay**. Sporozoites were dissected as described above and purified using 17% Accudenz protocol[71], where dissected sporozoites were placed on top of a 17% Accudenz solution and were centrifuged for 20 min at $2500 \times g$ at room temperature. In all, 1 mL of the top fraction was removed and centrifuged at $20,000 \times g$ for 10 min. Pelleted sporozoites were resuspended in simple DMEM and counted. In all, $2 \times 10^5$ sporozoites were then transferred into a tube that would be kept on ice or at 37 °C, with or without 10% FCS. After 30 to 120 min of incubation, sporozoites were pelleted again by centrifuging at $20,000 \times g$ for 10 min. Supernatants was separated from the pelleted sporozoites and sporozoites were lysed with RIPA buffer with 1% SDS for 1 h on ice. RIPA buffer comprised 50 mM Tris-HCl (NZYTech), 150 mM NaCl (Sigma), 5 mM EDTA (Sigma), 10 mM NaF (Sigma), 1% Triton X-100 (USB Corporation, Cleveland, OH, USA), 0.5% sodium deoxycholate (AppliChem, Maryland Heights, MO, USA). Following lysis, NuPAGE loading dye (Thermo Fisher Scientific) and β-mercaptoethanol (to a final concentration of 10%, Sigma) were added to all samples. Samples were boiled for 15 min at 95 °C.

**WB analysis**. Samples were separated in 10% Acrylamide gels and transferred to 0.2 μm pore-sized Nitrocellulose membranes (Bio-Rad). Membranes were placed inside Falcon tubes and blocked with 5% skim milk (Nestle, Vevey, Switzerland) overnight at 4 °C, on a tube roller. Membranes were then incubated with the appropriate primary antibodies, diluted in 5% skim milk, for 1 hour at room temperature. Membranes were washed with 0.1% Tween 20 (Sigma) in PBS, three times for 10 minutes. Membranes were then incubated with the appropriate secondary antibodies, conjugated to Horseradish peroxidase diluted in 5% skim milk, for 1 hour at room temperature, after which they were washed again three times for 10 minutes. WB were revealed using Amersham ECL Prime Blotting Detection Reagent (GE Healthcare) on the ChemiDoc XRS + system (Bio-Rad) using ImageLab software (v5.2.1). The primary antibodies used were rabbit αHA tag (1:1000 dilution, clone C29F4, Cell Signaling Technology, Danvers, MA, USA); mouse α*Pb*CSP (1:5000 dilution, clone 3D11, MR4) and rabbit α*Pb*BiP (1:1000 dilution, GenScript Biotech. Corp, New Jersey, NJ, USA). The secondary antibodies used were goat anti-mouse IgG F(ab')2, polyclonal antibody horseradish peroxidase (HRP) conjugate (1:5000 dilution, Enzo Life Sciences, Lausen, Switzerland); anti-rabbit IgG, HRP-linked antibody (1:3000 dilution, Cell Signaling Technology).

To assess the amount of secreted EXP2 protein, the intensity of the EXP2 band in the supernatant fraction was normalized to the intensity of the CSP band intensity at the respective time point in the pellet fraction. These values were further normalized to the ratio of EXP2/CSP in the pellet fraction at time zero (no stimulation).

All membranes used for this manuscript are presented in Supplementary Fig. 4.

**Acid sphingomyelinase knockdown**. Acid sphingomyelinase (gene name SMPD1) and Scramble short-hairpin RNAs were obtained from Mission shRNA (Sigma). In all, 800 ng/well of shRNAs were mixed with OptiMEM (Thermo Fisher Scientific) containing 0.3% Lipofectamine RNAiMax (Thermo Fisher Scientific) and added to 100,000 HepG2 cells. Cells were infected with sporozoites 48 hours after transfection. Samples were fixed 2 hours after infection and were stained as described above.

To assess the amount of knockdown of acid sphingomyelinase, mRNA was collected from uninfected cells, lysed at the time of infection. To quantify the amount of acid sphingomyelinase RNA, RNA was extracted using NZY Total RNA Isolation Kit, following manufacturer's instructions. 1 μg of extracted RNA was converted into cDNA using NZY First-Strand cDNA Synthesis Kit (NZYTech), as per manufacturer's instructions. cDNA was then used for qPCR, by measuring the abundance of *HuSMPD1* RNA, compared with *HuHPRT* RNA using either ViiA 7 (384-well plates, using QuantStudio v1.3 software) or 7500Fast (96-well plates, using 7500Fast v2.3 software) Real-Time PCR Systems (Thermo Fisher Scientific) using iTaq Universal SYBR Green Supermix (Bio-Rad Laboratories, Hercules, CA, USA). Analysis of results was performed using the $\Delta\Delta C_T$ method described above.

Primers used to assess aSMase knockdown:
*HsHPRT*– forward primer: TTTGCTGACCTGCTGGATTAC
*HsHPRT*– reverse primer: CAAGACATTCTTTCCAGTTAAAGTTG
*HuSMPD1* – forward primer: GGCCCACATTTGGGAAAGTT

*HuSMPD1* – reverse primer: TTCACCGGATGATCTTGCCT

**Electron microscopy**. Immuno-electron microscopy of purified parasites was performed according to Tokuyasu technique. In brief, the pellet was chemically fixed in 0.1 M phosphate buffer containing 2% formaldehyde and 0.2% glutaraldehyde, embedded in food-grade gelatine and cryopreserved in 2.3 M sucrose. Gelatine blocks were shaped in cubes and froze in liquid nitrogen and sectioned at −110 °C using a cryo-ultramicrotome (UC7 and FC7, Leica) to generate 70 nm sections. Sections were collected and thawed in a mixture of 2.3 M sucrose and 2% methylcellulose. Immuno-labeling was done in 1% bovine serum albumin and 0.8% gelatine from cold water fish skin in PBS with rabbit αEXP2 primary antibody (1:500 dilution) and 15 nm gold coupled Protein A (CMC Utrecht, The Netherlands, 1:50 dilution). After immuno-labeling, the sections were stained and mounted in a mixture of 3% (aq.) uranyl acetate and 2% methylcellulose. Images were recorded using a Hitachi H-7650 electron microscope (Hitachi, Tokyo, Japan) at 100 kV acceleration.

**Image analysis**. All immunofluorescence images were processed using FIJI software (version 1.52i) and macros written for each analysis to automate it. For WB images, the software ImageLab was used (version 5.2.1)

**Flow cytometry analysis**. Flow cytometry data were analyzed using FlowJo version X software (FlowJo LLC, Ashland, OR, USA).

**Statistical analysis**. Statistical analysis was performed using GraphPad Prism 5 software (GraphPad, La Jolla, CA). Mann–Whitney $U$ test was used to assess significance of differences observed between two groups and non-linear regression was performed to assess Desipramine inhibitory effects on invasion. Raw data and a description of statistical tests used is provided in Supplementary Information. Plots and figures were prepared using Adobe Illustrator (version CS4).

**Reporting summary**. Further information on research design is available in the Nature Research Reporting Summary linked to this article.

## Data availability

Replicates of WBs are provided in Supplementary Fig. 4. All other data are available from the authors upon reasonable request. Source data are provided with this paper.

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

## Acknowledgements

We thank Ana Parreira for producing *P. berghei*-infected Anopheles mosquitoes, Andreia Pinto for the Electron Microscopy work, the Rodent, Flow Cytometry and (especially)

Bioimaging Facilities of iMM JLA, Masafumi Yodha for the recombinant EXP2 plasmid, Mafalda Pimentel for the Gcamp6f plasmid, Joana Tavares for the αPbTRAP antibody, Maryse Lebrun and Margarida Ruivo for the αPbRON4 antibody and Paul Gilson for the RbαPbEXP2 antibody. Funding: This work was supported by grants from the Fundação para a Ciência e Tecnologia (PTDC/BIM-MEC/1342/2014 to M.M.M.) and Institut Mérieux (MRG_20052016 to M.M.M) V.Z.L. and J.M.-V. were sponsored by FCT fellowships (SFRH/BPD/81953/2011 and SFRH/BD/52226/2013, respectively).

## Author contributions

Conceptualization: J.M.-V., T.d.K-W., V.Z.-L., M.M.M.; investigation: J.M.-V.; resources: F.J.E., T.d.K-W.; writing—original draft: J.M.-V., T.d.K-W., V.Z.-L., M.M.M.; writing—review and editing: J.M.-V., F.J.E., T.d.K-W., V.Z.-L., M.M.M.; supervision: T.dK-W., V.Z.-L., M.M.M.; funding acquisition: V.Z.-L., M.M.M.

## Competing interests

The authors declare no competing interests.
