## [Peer Review File · Nature Communications]

Reviewer #1 (Remarks to the Author):

Review of Plasmodium EXP2 facilitates hepatocyte invasion by Mello-Vieira et al.

This is an intriguing paper, challenging but nevertheless will be of interest to others.

The authors explore the function of the parasite protein EXP2 in liver stages of infection. The protein, long implicated in the *Plasmodium* translocon as the functional pore (recently structurally resolved by Ho et al) has been shown to be essential for blood stages (in *P. falciparum*) and for progression through liver stages to blood (in *P. berghei*), however, its role at the point of colonization of the liver has not been explored. The authors use a previously published FLP/FRT system to conditionally excise the *exp2* gene prior to liver stages of infection, giving rise to ~50% loss of the gene, to then explore the effect on hepatocyte invasion. Remarkably, the authors show that a reduction in invasion seen can be complemented by recombinant EXP2 or a structurally/functionally analogous protein alpha-Hemolysin – suggesting pore formation is required for successful entry (though I am still unclear about when entry is – see below). A similar complementation of EXP2 reduced-entry phenotype can be achieved via addition of acid SMase – suggesting that pore formation, ion flux, upregulation of host-lipid pathway activation are intrinsic host-cell processes involved in liver infection by the parasite – something quite different to that envisaged for blood stages. Summarising the findings, the combined genetic, imaging and recombinant protein complementation work portrays a very different role for EXP2 in liver stages, potentially radically different from its function in the much better understood blood stages (or suggesting our understanding there is incomplete).

Overall the work is of a high quality and will be widely read and discussed. The challenge it faces is that the model “feels” inconsistent with some basic presumptions I would have had about liver stage infection (or indeed any stages of infection by *Plasmodium* parasites). This does not mean I think it is wrong, only that there are gaps in the data that still leave me feeling uneasy about whether the interpretation is robust or not. As a hypothesis, however, I think the work will stimulate much thinking and work in the area, and I would therefore encourage publication (I am in favour of work that stimulates discussion/further work). However, I think it is important that the current work is portrayed in this light, as a model at present, suggestive of this new role for EXP2 but with some key caveats/challenges (which constitute my main concerns with the work).

Major comments:

1. **cKO is not complete**: It is a shame that excision isn't 100%. Dealing with a phenotype that is only applicable to 45% of parasites in a population reduces the statistical power of observations made, lessening the impact and leads to some inevitable lingering doubts/questions. E.g. whilst gene excision is clear for 45% of parasites (by qPCR), I find the imaging (IFA) less convincing as a quick adjustment of image levels (e.g. by photoshop) clearly shows that in this image there is something in the green channel... is all protein gone? Was residual mRNA left over from pre-excision that means some protein is present? I wouldn't ordinarily mind this, but this issue has been an ongoing battle in zoite invasion literature (sometimes quite heated - e.g. see examples on AMA1, RONs, myosin and actin!). Here the issue is less problematic for the central narrative of the paper since there is a phenotype even with 50% loss, but where there is no phenotype (e.g. long-term development post-invasion) I am less convinced. E.g. is EXP2 really not essential for post-invasion development? Don't they still have a translocon (i.e. protein export), how on earth could development proceed without export and the PTEX pore? I do not feel this latter point is adequately addressed.

2. **EXP2 localisation is still most likely within the dense granules (DGs) which are expected to be released AFTER invasion**: I am calling invasion anything up to junction sealing, post-invasion then follows. My understanding from the blood stages field is that EXP2 is a dense granule protein (e.g. Bullen et al). Looking at the immuno-EM and IFA data (Fig. 2a-b) I think this remains the most parsimonious localisation for it in sporozoite stages though the quality of both EM and IFA are not informative enough to conclude anything - gold particles are quite evenly distributed and IFAs are not clear. Assuming localisation is in the DGs (how could it be in different places in two different stages?) is however problematic. My expectation would be that DG release is POST

invasion (after junction sealing), not before (evidence from *Toxoplasma* and merozoites). Thus, the expectation, as with blood stages, is that EXP2 function would not precede invasion but would be after invasion has completed (immediately post-invasion) – e.g. set up of the translocon (Riglar et al). Whilst the sporozoite secretion assays (Fig. 2c) are convincing of secretion yes, they don't prove pre-invasion (as this could happen by premature activation – something you can trigger in *Toxoplasma* too). So, the timing of EXP2 secretion and interaction with the host cell becomes a critical, yet to my mind unresolved, question here. It would be critically at odds with the complementation work where addition of rEXP2 at invasion seems to rescue development. If in normal circumstances EXP2 is only released just post-invasion (junction is sealed), then the model presented cannot be correct. I do not feel this dichotomy is adequately addressed in the evidence presented (i.e. resolving precisely PRE vs. POST invasion release).

Incidentally (a small point), I also find the discussion about down regulation of mRNA transcripts post-invasion (Fig 2e-f) a bit unconvincing. The relative ratios are very slight <1.5 fold different to control (up or down depending on gene), which to my mind could be within the realm of noise... Much more would need to be done to really resolve the absolute expression of invasion proteins pre and post-invasion. That EXP2 protein is not found post-invasion (quickly dissipates) would be entirely consistent with DG release (as it is in blood stages). DG release really is the first thing that happens!

3. **The timing of events in ex vivo experiments should be discussed with caution.** The idea of a 1 hour pause, even though the authors are not the first to propose this, I find confusing. In reality, invasion happens in deep vasculature, the sporozoite having navigated from the skin through the blood stream to the sinusoid. That it would then wait an hour (post-traversal even) seems unlikely. Instead, this is most likely a phenomenon of in vitro invasion assays where we are going from salivary gland dissection straight to liver cells – missing out the bite, the skin, the blood stream and the sinusoid... So I think the authors should treat any observations about a pause with caution – since in reality this isn't likely a natural phenomenon.

4. **Could exogenous pore-formation still lead to a phenotype?** Doubting a role for EXP2 at invasion (favouring a post-invasion hypothesis myself), the most striking observation of the paper is then the effect of exogenous rEXP2 or alpha-Hemolysin, and recombinant acid SMase, and the ability of each to rescue the invasion defect phenotype (plus supporting ion work Fig. 3). I have no explanation for why these would complement an entry/early development defect beyond invasion, i.e. the conclusion from the authors, that by inducing pore formation, there's some upregulation of a signalling pathway that facilitates invasion further. So I am left with the question, could this be an unnatural phenomenon? Unrelated to EXP2's native role? I think that can't be ruled out, although that sounds perhaps more difficult to imagine than the role posited by the authors. Again, this is where presentation of a model should be supported, but with caveats of possible alternative hypotheses.

So, if I had to summarise my disagreement, it would be with two sentences:

"Altogether, our data show that *Plasmodium* sporozoites **secrete EXP2 upon contact with host cell milieu**, which leads to the formation of pores in the hepatocyte plasma membrane and the activation of the membrane repair pathway, the latter of which is hijacked by the parasite to facilitate invasion of the host cell." And...

"...we propose that **EXP2 is secreted from the sporozoite to the hepatocyte membrane**, creating pores that generate ion fluxes, which in turn trigger the release of host acid SMase and induces endocytosis, facilitating parasite invasion"

If we are talking about invasion (contact through to junction formation and junction sealing) I don't see how a dense granule protein could play that role and I am left confused by how EXP2 could do such potentially different things in two different stages. Nature normally chooses an easy path.

However, if we are talking about infection (junction closure through to intra-hepatocyte development) precisely when dense granule release would be expected, then there would be no

conflict with data from blood stages and the challenge would be explaining the exogenous rEXP2 experiments... Which I don't have an answer for!

Some additional thoughts (with no expectation of these being done, especially with a global pandemic on):

1. Have you thought to look at EXP2 during sporozoite invasion (i.e. in relation to RON complex)?
2. Have you tried live video microscopy of invasion with cKO EXP2 parasites? I know this isn't easy but something like the work from Gonzalez et al. Might help see where the defect occurs.
3. Could intra-vital microscopy help to see if cKO EXP2 parasites even make it to the liver/enter the sinusoid/enter hepatocytes (ridiculously challenging I know).

Reviewer #2 (Remarks to the Author):

This is a fascinating paper with evidence that Exp2 is critical for invasion of sporozoites into liver cells. The interesting story is the comparison to asexual parasites where it plays a critical role in proteins entering the host cell from the parasitophorous vacuolar membrane and for the entry of small molecules from the red cell into the parasite. The critical difference is that one of the components of the translocon, HSP101, is lacking in sporozoites. Secondly, its site of action is at the liver stage membrane for invasion and not for its function in modifying the vacuole for delivery of proteins and in delivering small molecules to the parasite. I think that the paper would be improved by a little more description of the role in the asexual infected red cell. They do go into it, but I think a little more information on it in the discussion would help those who know little about this critical molecule in the malarial parasite.

Now for the evidence: The use a conditional knockdown to demonstrate its effect. It is too bad that they didn't use the one from the Blackman lab that causes a greater loss up to 98%. The advantage of the partial KO is that they could measure the activity in sporozoites that did invade and show Exp2 activity in those that invade. The stronger KO would have improved the liver experiments with Exp2 added after adding sporozoites for its effect on invasion.

Where is Exp2 in sporozoites? They are wise to not try to define its location because of the problems with immunoEM. The use of other markers to show it is not in certain organelles is a problem because in the asexual parasite both AMA1 and EBA175 are found in micronemes, but never in the same microneme. Importantly, they show that Exp2 ends up in the supernatant around stimulated sporozoites as evidence Exp2 is in a secretory organelle. In asexual parasites, it is thought to be in the dense granules and released into the space around the parasite during invasion. Again, it should be noted that its exact localization in schizonts is difficult to be sure about.

What is its function? First, they show an increase of calcium uptake when the hepatocyte is exposed to PfExp2. Next they knock down liver acid sphingomyelinase and show its effect on invasion. These studies are interesting for their importance in giving evidence for the mechanism, but may have problems in direct proof of the mechanism.

Overall, an interesting story that is unique and should stimulate much research on the mechanism of sporozoite invasion and opens a whole new area in sporozoite invasion.

Louis H. Miller

Reviewer #3 (Remarks to the Author):

This manuscript by Mello-Vieira et al reports a novel function for EXP2 in hepatocyte invasion by Plasmodium berghei sporozoites. EXP2 is the pore-forming component of the PTEX translocon and is known to be important for protein export and small molecule transport across the parasite vacuolar membrane during the blood-stage. However, the function of EXP2 at other stages of the parasite life cycle is unclear. Previously, a conditional excision mutant of EXP2 was generated with the FLP/FRT system and found to result in EXP2 knockdown in ~50% of sporozoites. This partial loss of EXP2 impacts the transition from sporozoites through the parasite liver stage and into the blood stage. However, the basis for the importance of EXP2 during these stage transitions remains

unknown. Here, using the same conditional mutant, the authors find that EXP2 is specifically important for sporozoite infection of hepatocytes. EXP2 expression is found to be upregulated and secreted upon sporozoite activation, in coordination with invasion timing. Addition of recombinant EXP2, the pore forming protein alpha-hemolysin and acid sphingomyelinase can rescue the invasion defect while desipramine, a pharmacological inhibitor of aSMase, antagonizes invasion. Collectively, these results lead the authors to propose a novel role for EXP2 in hepatocyte invasion through host membrane wounding resulting in activation of host membrane repair pathways that involve the exocytosis of aSMase, a process known to be exploited by other intracellular pathogens. This is an interesting study that reveals an unexpected additional function for EXP2 and has important implications for the host cell contribution to sporozoite invasion. However, I have some concerns about transparency with some of the experiments and reagents. My specific comments are below.

-I'm confused about the data displayed in Fig 1e. This is a comparison of the qPCR data in 1a (n=4) and the invasion analysis by IFA in Fig 1d (n=12). How does n=10? Are these the same experiments shown in 1a or additional replicates? If there are more than the 4 replicates for the qPCR shown in 1a, why have these data been excluded from 1a? As such, it appears that 1a does not capture the full range of the data observed as shown in 1e (~100% WT locus/infection to ~25% WT locus/infection). Also, in 1e why is there a mismatch between the n for qPCR in 1e (10) and the n for the IFA invasion assay (30)? Is n=30 for the IFA assay 10 biological replicates with 3 technical replicates each, or 30 biological replicates?

-Lines 252-254: The authors suggest the failure of rescue when rEXP2 is added at time 0 is due to a failure to coordinate with expression/secretion of other factors involved in invasion. Is the rEXP2 turnover in the culture (by binding/uptake by host cells, degradation, etc) sufficiently rapid to no longer be available after 1 hour? This could be tested with a western blot on the culture supernatant at time the of addition and after 60 min incubation.

-In the Gcamp6f and GEPII experiments shown in Fig 3c and d, the concentration of rEXP2 used is 10 times that of the highest concentration in the invasion rescue assays. Can these effects be detected at the level of EXP2 treatment used in the rescue assays? Does a 100 nM treatment still rescue invasion? I appreciate that effective concentrations may differ between these experiments with very different readouts, but if so this should be discussed.

-Please show P values for all comparisons, even if they fall above a significance threshold. For example, it is important to state the p value comparing the WT and EXP2 cKO at the 60 min time point in Fig 2d to make the argument that the treatment is rescuing. Same point for the aSMase treatment comparison in Fig 3g, etc.

-It is not always clear which EXP2 antibody is being used in which experiment. For example, in the IFAs in Fig 2b or Sup Fig 2c, it is not clear if this is the rabbit or the mouse anti-PfEXP2. Also, no information is provided in the methods about the RON4 antibody. References should be provided for all non-commercial antibodies if they have been previously published. If they have not been published, information about their generation and validation should be provided. Given that both anti-EXP2 antibodies were raised against *P. falciparum* EXP2, it is important to validate that they cross react with *Pb*EXP2. Is the rabbit anti-PfEXP2 used here the same as that used in Matthews et al 2013?

-Why are EXP2-HA parasites used for secretion assays while untagged lines are used for other experiments related to EXP2 secretion during sporozoite invasion (for example, the IFAs in Fig 2b and Sup Fig 2c)? This crucial experiment seems to be the only place this HA tagged line is used. Do the EXP2 antibodies not work in western blots?

-In fig 2e, expression levels of GAP45 and EXP1 are used to argue that the pattern seen for EXP2 is unique (peaking at 60 minutes). It is claimed from the data that GAP45 is progressively downregulated while EXP1 is progressively upregulated. Are the differences between the time points for GAP45 or EXP1 significant?

Minor comments:

-line 277: The reference given is #21 but should be #26.

-The methods refer to a number of assays (calcium-free invasion experiments, lines 382-384; ATP depletion experiments, lines 507-509) that do not appear to be present in the manuscript.

-lines 760-762: The legend mentions the blue shading corresponding to the UIS4 cKO but not the green shading that corresponds to the TRAP cKO.

-line 855: knock should be knockdown.

-line 934: There is no figure 2g or j. The authors are referring to 2c and f.

Point-by-point response to the reviewers

We want to thank the reviewers for the time and effort dedicated to offer valuable feedback on the manuscript. We have now responded to all the insightful comments and incorporated changes in the manuscript to reflect the provided suggestions. All the changes made have been highlighted within the manuscript.

Reviewer #1:

This is an intriguing paper, challenging but nevertheless will be of interest to others.

The authors explore the function of the parasite protein EXP2 in liver stages of infection. The protein, long implicated in the *Plasmodium* translocon as the functional pore (recently structurally resolved by Ho et al) has been shown to be essential for blood stages (in *P. falciparum*) and for progression through liver stages to blood (in *P. berghei*), however, its role at the point of colonization of the liver has not been explored. The authors use a previously published FLP/FRT system to conditionally excise the *exp2* gene prior to liver stages of infection, giving rise to ~50% loss of the gene, to then explore the effect on hepatocyte invasion. Remarkably, the authors show that a reduction in invasion seen can be complemented by recombinant EXP2 or a structurally/functionally analogous protein alpha-Hemolysin – suggesting pore formation is required for successful entry (though I am still unclear about when entry is - see below). A similar complementation of EXP2 reduced-entry phenotype can be achieved via addition of acid SMase – suggesting that pore formation, ion flux, upregulation of host-lipid pathway activation are intrinsic host-cell processes involved in liver infection by the parasite – something quite different to that envisaged for blood stages. Summarising the findings, the combined genetic, imaging and recombinant protein complementation work portrays a very different role for EXP2 in liver stages, potentially radically different from its function in the much better understood blood stages (or suggesting our understanding there is incomplete).

Overall the work is of a high quality and will be widely read and discussed. The challenge it faces is that the model “feels” inconsistent with some basic presumptions I would have had about liver stage infection (or indeed any stages of infection by *Plasmodium* parasites). This does not mean I think it is wrong, only that there are gaps in the data that still leave me feeling uneasy about whether the interpretation is robust or not. As a hypothesis, however, I think the work will stimulate much thinking and work in the area, and I would therefore encourage publication (I am in favour of work that stimulates discussion/further work). However, I think it is important that the current work is portrayed in this light, as a model at present, suggestive of this new role for EXP2 but with some key caveats/challenges (which constitute my main concerns with the work).

We truly thank the reviewer for the helpful suggestions and criticisms. We appreciate her/his effort to improve our manuscript and believe we have now done so, namely, by interpreting data through different premises and perspectives.

Major comments:

1. **ckO is not complete:** It is a shame that excision isn't 100%. Dealing with a phenotype that is only applicable to 45% of parasites in a population reduces the statistical power of observations made, lessening the impact and leads to some inevitable lingering doubts/questions. E.g. whilst gene excision is clear for 45% of parasites (by qPCR), I find the imaging (IFA) less convincing as a quick adjustment of image levels (e.g. by photoshop) clearly shows that in this image there is something in the green channel... is all protein gone? Was residual mRNA left over from pre-excision that means some protein is present? I wouldn't ordinarily mind this, but this issue has been an ongoing battle in zoite invasion literature (sometimes quite heated - e.g. see examples on AMA1, RONs, myosin and actin!). Here the issue is less problematic for the central narrative of the paper since there is a phenotype even with 50% loss, but where there is no phenotype (e.g. long-term development post-invasion) I am less convinced. E.g. is EXP2 really not essential for post-invasion development? Don't they still have a translocon (i.e. protein export), how on earth could development proceed without export and the PTEX pore? I do not feel this latter point is adequately addressed.

We totally agree with the reviewer that not having a full KO is not ideal but despite this we were able to demonstrate a phenotype that correlated with the degree of excision. We understand the reviewers' concern with the conclusion of all parasites by 48h being KO that there was no effect on development, a fact that also surprised us.

One possible explanation for this effect is that the canonical translocon system in the blood stages might not be working during the liver stages of infection or that protein export into a very different type of host cell occurs by another means. In fact, not all the components of the translocon are expressed during the liver stage, namely, the protein HSP101 (the motor of the translocon) is absent from the liver stage PVM¹. Moreover, Ming Kalanon and colleagues showed that a modified version of the KAHRP protein when fused to GFP (which is exported during the blood stage and requires HSP101 for unfolding) was not exported to the hepatocyte cytosol during the liver stage of infection². These two observations suggest that EXP2 might serve a different function to protein transport during the liver stage. This does not mean that the parasite does not possess a system for protein export, it just might not rely on the machinery used during the BS. For example, *Toxoplasma gondii*, which possess an EXP2 orthologue³, utilizes another channel protein to export its proteins into nucleated host cells, so far unknown³. We have added our interpretation to these results in the results/discussion section of the manuscript.

2. **EXP2 localisation is still most likely within the dense granules (DGs) which are expected to be released AFTER invasion:** I am calling invasion anything up to junction sealing, post-invasion then follows. My understanding from the blood stages field is that EXP2 is a dense granule protein (e.g. Bullen et al). Looking at the immuno-EM and IFA data (Fig. 2a-b) I think this remains the most parsimonious localisation for it in sporozoite stages though the quality of both EM and IFA are not informative enough to conclude anything - gold particles are quite evenly distributed and IFAs are not clear. Assuming localisation is in the DGs (how could it be in different places in two different stages?) is however problematic. My expectation would be that DG release is POST invasion (after junction sealing), not before (evidence from

Toxoplasma and merozoites). Thus, the expectation, as with blood stages, is that EXP2 function would not precede invasion but would be after invasion has completed (immediately post-invasion) – e.g. set up of the translocon (Riglar et al). Whilst the sporozoite secretion assays (Fig. 2c) are convincing of secretion yes, they don't prove pre-invasion (as this could happen by premature activation – something you can trigger in Toxoplasma too). So, the timing of EXP2 secretion and interaction with the host cell becomes a critical, yet to my mind unresolved, question here. It would be critically at odds with the complementation work where addition of rEXP2 at invasion seems to rescue development. If in normal circumstances EXP2 is only released just post-invasion (junction is sealed), then the model presented cannot be correct. I do not feel this dichotomy is adequately addressed in the evidence presented (i.e. resolving precisely PRE vs. POST invasion release).

Incidentally (a small point), I also find the discussion about down regulation of mRNA transcripts post-invasion (Fig 2e-f) a bit unconvincing. The relative ratios are very slight <1.5 fold different to control (up or down depending on gene), which to my mind could be within the realm of noise... Much more would need to be done to really resolve the absolute expression of invasion proteins pre and post-invasion. That EXP2 protein is not found post-invasion (quickly dissipates) would be entirely consistent with DG release (as it is in blood stages). DG release really is the first thing that happens!

The reviewer raises an important point. Given that we stimulate sporozoites with a pungent treatment, we might be inducing the release of all vesicular proteins. As such, we lose the ability to assess the timing of release, between pre-invasion (micronemes/rhoptries) and post-invasion (dense granules).

EXP2 is present in the dense granules in merozoites⁴. However, it is not clear where is EXP2 present during the sporozoite stage. Using Electron Microscopy, we could not locate it to the rhoptries. As such, we are left with micronemes and dense granules. Dense granules have not been identified yet in the sporozoite stage but based on data from merozoites and other apicomplexan parasites, such as *Toxoplasma gondii*, they are thought to be present.

Importantly, due to the traversal phenomena, the sporozoite might have more secretory vesicles than the merozoite stage. In fact, CelTOS is discharged from the sporozoite in order for it to traverse hepatocytes and has been located to the micronemes⁵. In that sense, it is possible that sporozoites possess two types of micronemes, one for traversal (released early) and one for initiating invasion (released only after activation – where we think EXP2 might be stored).

We cannot exclude other roles for EXP2 during the post-invasion process, especially as a dense granule protein, EXP2 might be incorporated in the nascent PVM, not being required for the entry into the hepatocyte but for vacuole integrity and parasite maturation during the few minutes/hours after the parasite has sealed the PVM. This requirement for EXP2 has been observed for merozoites that lack the RON3 protein. RON3-deleted merozoites can invade erythrocytes but cannot develop further, because EXP2 cannot exert its functions (both protein translocation and solute channel)⁶.

In the liver stages, EXP2 might serve a similar role, being required for the maturation of sporozoites into EEFs. For instance, EXP2 might be needed to change the liver stage vacuole from transient into definitive, as suggested by the work of Risco-Castillo⁷.

Notably, our results with the EXP2 cKO and the in/out staining (Figure 1i) suggest that parasites lacking EXP2 are not even able to enter cells, as we observe them on the extracellular milieu. This, in accordance with our rescue experiments (Figure 2d and Figure

3a) also suggest that sporozoites might need EXP2 to start the invasion process, rather than finish it.

One important consideration is that EXP2 might create pores at the surface of the sporozoite, leading to the increase of intracellular calcium, triggering the release of more invasion proteins. This EXP2-induced calcium increase would need to be timed with the expression of other invasion proteins, as the addition of rEXP2 to unactivated sporozoites does not rescue the invasion capacity of EXP2 cKO. In fact, in our rescue experiments, we do not know where EXP2 is acting. It can be acting on the sporozoite in an autocrine fashion or in the hepatocyte in a paracrine way.

We have added these ideas in the discussion section of the manuscript and updated the final version of the model to take these observations and hypotheses into account (Fig. 4, please see **Figure R1**).

Figure R1 – New model for the role of EXP2 during the invasion of Plasmodium sporozoites.

3. The timing of events in ex vivo experiments should be discussed with caution. The idea of a 1 hour pause, even though the authors are not the first to propose this, I find confusing. In reality, invasion happens in deep vasculature, the sporozoite having navigated from the skin through the blood stream to the sinusoid. That it would then wait an hour (post-traversal even) seems unlikely. Instead, this is most likely a phenomenon of in vitro invasion assays where we are going from salivary gland dissection straight to liver cells – missing out the bite, the skin, the blood stream and the sinusoid... So I think the authors should treat any observations about a pause with caution – since in reality this isn't likely a natural phenomenon.

We thank to reviewer for pointing this out. Indeed, we are comparing an *in vitro* event to a several *in vivo* phenomena and we were not clear in explaining the “pause hypothesis”.

As the reviewer mentions, the one-hour pause has been observed for *in vitro* conditions before. It is possible that *in vivo* this timing is different, as the conditions that the sporozoite is exposed to are different between cell culture and a mammalian dermis/circulatory system. However, we think the concept of the pause also occurs *in vivo*. For instance, when observing the injection site after a mosquito bite, Yamauchi and colleagues observed that parasites linger in the injected site for more than 1 hour⁸. Indeed, the time spent by the sporozoite in the skin has lead several authors to establish the idea that there is a “skin phase of the infection”, as it is estimated that sporozoites spend more time in the skin than migrating to the liver and invading an hepatocyte⁹. As such, it is conceivable that the sporozoite has evolved a mechanism to avoid skin infection, by delaying the expression of invasion proteins and prioritizing the gliding machinery, to reach circulation. Once in circulation, we think that the sporozoite would sense the difference in environment (nutrients, serum, flow) and that this would trigger changes in the sporozoite proteome.

To be clearer, we are proposing that, as previously described, the pause happens before the parasites reach the capillaries and the circulatory system, and that this mechanism avoids skin cell invasion, pausing the expression of invasion-related proteins, until it reaches the liver.

We have added this clarification to the discussion section of the manuscript.

4. Could exogenous pore-formation still lead to a phenotype? Doubting a role for EXP2 at invasion (favouring a post-invasion hypothesis myself), the most striking observation of the paper is then the effect of exogenous rEXP2 or alpha-Hemolysin, and recombinant acid SMase, and the ability of each to rescue the invasion defect phenotype (plus supporting ion work Fig. 3). I have no explanation for why these would complement an entry/early development defect beyond invasion, i.e. the conclusion from the authors, that by inducing pore formation, there's some upregulation of a signalling pathway that facilitates invasion further. So I am left with the question, could this be an unnatural phenomenon? Unrelated to EXP2's native role? I think that can't be ruled out, although that sounds perhaps more difficult to imagine than the role posited by the authors. Again, this is where presentation of a model should be supported, but with caveats of possible alternative hypotheses.

The reviewer gives us an intriguing task. As above (in point 2), how to reconcile our observations with an independent action for EXP2 (and hemolysin and aSMase).

Recently, it was described that after invasion of hepatocytes, *Plasmodium* parasites mature from sporozoites to EEFs after a calcium influx in the host cell, that is dependent on the action

of PKC ζ ¹⁰. It is conceivable that rPfEXP2 and/or α -hemolysin are triggering the host membrane repair pathway (via an initial calcium influx), that activates aSMase, which lastly activates PKC ζ , inducing another and more intense calcium influx. This could mean that all the manipulations that we are doing on the host cell are only contributing to the calcium influx described by Bando and colleagues¹⁰.

Addressing this in light of a complete invasion mechanism, it would mean that secreted EXP2 might have an effect on the host cell, not for invasion but for the transformation of sporozoite into EEF.

So, if I had to summarise my disagreement, it would be with two sentences:

“Altogether, our data show that Plasmodium sporozoites **secrete EXP2 upon contact with host cell milieu**, which leads to the formation of pores in the hepatocyte plasma membrane and the activation of the membrane repair pathway, the latter of which is hijacked by the parasite to facilitate invasion of the host cell.” And...

“...we propose that **EXP2 is secreted from the sporozoite to the hepatocyte membrane**, creating pores that generate ion fluxes, which in turn trigger the release of host acid SMase and induces endocytosis, facilitating parasite invasion”

If we are talking about invasion (contact through to junction formation and junction sealing) I don't see how a dense granule protein could play that role and I am left confused by how EXP2 could do such potentially different things in two different stages. Nature normally chooses an easy path.

However, if we are talking about infection (junction closure through to intra-hepatocyte development) precisely when dense granule release would be expected, then there would be no conflict with data from blood stages and the challenge would be explaining the exogenous rEXP2 experiments... Which I don't have an answer for!

We agree with the reviewer that the sentences above are too persuasive for the experimental results we obtained. As such, we have changed them, to highlight what are our hypothesis and the actual experimental results.

“Altogether, our data show that *Plasmodium* sporozoites secrete EXP2 upon stimulation and that recombinant EXP2 can lead to the formation of pores in the hepatocyte plasma membrane. Both exogenous EXP2 and acid SMase can rescue the invasion impairment of EXP2 cKO sporozoites, suggesting that sporozoites hijack the host membrane repair pathway, to facilitate invasion of the host cell.”

And

“For the invasion of hepatocytes, one hypothesis is that EXP2 is secreted from the sporozoite to the hepatocyte membrane, creating pores that generate ion fluxes, which in turn trigger

the release of host acid SMase and induces endocytosis, facilitating parasite invasion (Fig. 4). [...]

On the other hand, our sporozoite secretion assay does not allow us to distinguish between pre-invasion and post-invasion release. As such, it is also possible that EXP2 might be incorporated in the nascent PVM, not being required for the invasion process itself but for vacuole integrity and parasite maturation during the few minutes/hours after the parasite has sealed the PVM, similarly to what happens during the blood stage of *Plasmodium* infection.”

We have also changed the model (Fig. 4, please see **Figure R1**) to accommodate different possible explanations.

Some additional thoughts (with no expectation of these being done, especially with a global pandemic on):

1. Have you thought to look at EXP2 **during** sporozoite invasion (i.e. in relation to RON complex)?
2. Have you tried live video microscopy of invasion with cKO EXP2 parasites? I know this isn't easy but something like the work from Gonzalez et al. Might help see where the defect occurs.
3. Could intra-vital microscopy help to see if cKO EXP2 parasites even make it to the liver/enter the sinusoid/enter hepatocytes (ridiculously challenging I know).

These are indeed very interesting but very challenging experiments.

Hepatocyte invasion rate is very low and we do not know when the parasite is invading or merely traversing a hepatocyte. To distinguish these processes we would need to perform the experiments using vacuolar markers, that are incompatible with the live imaging setting. Moreover, our EXP2 cKO parasite line does not express a fluorescent marker. We would require the generation of a new parasite line, which would be impractical with the time frame for this publication.

Reviewer #2:

This is a fascinating paper with evidence that Exp2 is critical for invasion of sporozoites into liver cells. The interesting story is the comparison to asexual parasites where it plays a critical role in proteins entering the host cell from the parasitophorous vacuolar membrane and for the entry of small molecules from the red cell into the parasite. The critical difference is that one of the components of the translocon, HSP101, is lacking in sporozoites. Secondly, its site of action is at the liver stage membrane for invasion and not for its function in modifying the vacuole for delivery of proteins and in delivering small molecules to the parasite. I think that the paper would be improved by a little more description of the role in the asexual infected red cell. They do go into it, but I think a little more information on it in the discussion would help those who know little about this critical molecule in the malarial parasite.

The EXP2 protein and the translocon complex are truly fascinating. We have added more information on the role of EXP2 both in the introduction and the discussion sections of the manuscript.

Now for the evidence: The use a conditional knockdown to demonstrate its effect. It is too bad that they didn't use the one from the Blackman lab that causes a greater loss up to 98%. The advantage of the partial KO is that they could measure the activity in sporozoites that did invade and show Exp2 activity in those that invade. The stronger KO would have improved the liver experiments with Exp2 added after adding sporozoites for its effect on invasion.

The reviewer concerns are justified, this conditional strategy does have its limitations. It would be great to use the DiCre-mediated gene excision system developed by Mike Blackman and collaborators¹¹. Unfortunately, this system has not yet be developed and adapted for *P. berghei*. We did take advantage of an imperfect system to show that the extent of excision, which was variable between experiments, correlated with a reduction in invasion capacity of the sporozoites.

Where is Exp2 in sporozoites? They are wise to not try to define its location because of the problems with immunoEM. The use of other markers to show it is not in certain organelles is a problem because in the asexual parasite both AMA1 and EBA175 are found in micronemes, but never in the same microneme. Importantly, they show that Exp2 ends up in the supernatant around stimulated sporozoites as evidence Exp2 is in a secretory organelle. In asexual parasites, it is thought to be in the dense granules and released into the space around the parasite during invasion. Again, it should be noted that its exact localization in schizonts is difficult to be sure about.

We thank the reviewer for this comment. Our initial idea was to assess the specific organelle in which EXP2 resides. Unfortunately, we could not get a definitive answer with the imaging techniques and protein markers we have used.

EXP2 is present in the dense granules in merozoites⁴. However, it is not clear where is EXP2 present during the sporozoite stage. Using Electron Microscopy, we could not locate it to the rhoptries. As such, we are left with micronemes and dense granules. In the model we propose for invasion, EXP2 would be better placed in the micronemes, as it could initiate the invasion process. However, we have no results can demonstrate that. As Reviewer 1 pointed out, it

might be present in the dense granules, as it is during the merozoite stage. Dense granules have not been identified yet in the sporozoite stage but based on data from merozoites and other apicomplexan parasites, such as *Toxoplasma gondii*, they are thought to be present. Importantly, due to the traversal phenomena, the sporozoite might have more secretory vesicles than the merozoite stage. In fact, CelTOS is discharged from the sporozoite in order for it to traverse hepatocytes and has been located to the micronemes⁵. In that sense, it is possible that sporozoites possess two types of micronemes, one for traversal (released early) and one for initiating invasion (released only after activation – where we think EXP2 might be stored).

What is its function? First, they show an increase of calcium uptake when the hepatocyte is exposed to PfExp2. Next they knock down liver acid sphingomyelinase and show its effect on invasion. These studies are interesting for their importance in giving evidence for the mechanism, but may have problems in direct proof of the mechanism. Overall, an interesting story that is unique and should stimulate much research on the mechanism of sporozoite invasion and opens a whole new area in sporozoite invasion.

We wish to thank the reviewer for the positive feedback. We have now fully addressed the points raised, which have certainly contributed to improve the clarity of the message.

Reviewer #3:

This manuscript by Mello-Vieira et al reports a novel function for EXP2 in hepatocyte invasion by *Plasmodium berghei* sporozoites. EXP2 is the pore-forming component of the PTEX translocon and is known to be important for protein export and small molecule transport across the parasite vacuolar membrane during the blood-stage. However, the function of EXP2 at other stages of the parasite life cycle is unclear. Previously, a conditional excision mutant of EXP2 was generated with the FLP/FRT system and found to result in EXP2 knockdown in ~50% of sporozoites. This partial loss of EXP2 impacts the transition from sporozoites through the parasite liver stage and into the blood stage. However, the basis for the importance of EXP2 during these stage transitions remains unknown. Here, using the same conditional mutant, the authors find that EXP2 is specifically important for sporozoite infection of hepatocytes. EXP2 expression is found to be upregulated and secreted upon sporozoite activation, in coordination with invasion timing. Addition of recombinant EXP2, the pore forming protein alpha-hemolysin and acid sphingomyelinase can rescue the invasion defect while desipramine, a pharmacological inhibitor of aSMase, antagonizes invasion. Collectively, these results lead the authors to propose a novel role for EXP2 in hepatocyte invasion through host membrane wounding resulting in activation of host membrane repair pathways that involve the exocytosis of aSMase, a process known to be exploited by other intracellular pathogens. This is an interesting study that reveals an unexpected additional function for EXP2 and has important implications for the host cell contribution to sporozoite invasion. However, I have some concerns about transparency with some of the experiments and reagents.

We want to thank the referee for the insightful comments, which we believe have helped us improve the quality of the manuscript.

My specific comments are below.

-I'm confused about the data displayed in Fig 1e. This is a comparison of the qPCR data in 1a (n=4) and the invasion analysis by IFA in Fig 1d (n=12). How does n=10? Are these the same experiments shown in 1a or additional replicates? If there are more than the 4 replicates for the qPCR shown in 1a, why have these data been excluded from 1a? As such, it appears that 1a does not capture the full range of the data observed as shown in 1e (~100% WT locus/infection to ~25% WT locus/infection). Also, in 1e why is there a mismatch between the n for qPCR in 1e (10) and the n for the IFA invasion assay (30)? Is n=30 for the IFA assay 10 biological replicates with 3 technical replicates each, or 30 biological replicates?

We apologize for the lack of clarity in the data report.

In Fig 1a, we present the result of 4 experiments (4 independent mosquito infections) where we assessed the excision rate of the EXP2 cKO system, using both the TRAP and UIS4 systems. In Fig 1d data corresponds to other 4 independent experiments (another set of 4 mosquito infections using only the UIS4-mediated excision of EXP2), analysed at three time points, each with three technical replicates. As such, we considered we analyzed 4 independent experiments with 12 total replicates for each timepoint. These were analyzed at 2 hours after infection in parallel with 24 and 48 hours after infection. This set of experiments were performed independently of the data presented in Fig 1a. In Fig 1e we have a another set of

10 biological replicates (10 new mosquito infections), independent from the experiments analyzed in Fig 1a and Fig 1d. For each of these 10 experiments, excision rate was analyzed by qPCR once (no other technical replicates) and invasion rate was assessed by IFA using 3 technical replicates. As such, we have 3 technical x 10 biological replicates. We have changed the legend of the figures to better explain the experimental setting.

Relating to the variability of the conditional KO system, we decided to include the value of 100% and 25% (our worst and best excision rate, respectively) to better illustrate the variability of the conditional KO system. This way, the data reflects the range of possible excision rates we encountered during these experiments.

-Lines 252-254: The authors suggest the failure of rescue when rEXP2 is added at time 0 is due to a failure to coordinate with expression/secretion of other factors involved in invasion. Is the rEXP2 turnover in the culture (by binding/uptake by host cells, degradation, etc) sufficiently rapid to no longer be available after 1 hour? This could be tested with a western blot on the culture supernatant at time the of addition and after 60 min incubation.

We thank the reviewer for this observation. If EXP2 is indeed acting as a pore-forming protein in the host cell membrane, we hypothesize that it would very quickly interact with and be incorporated/internalized in/by hepatocyte membranes. The work of Castro-Gomes and colleagues proposes that pore-forming proteins would be internalized by the cell in 90 seconds¹². As such, we think that there is no more protein available after 1 hour in contact with the cells.

When we treated HepG2 cells with 1 nM of *rPfEXP2*, we observe a decrease in *rPfEXP2* concentration in the supernatant already at 5min after cells are exposed to the protein. This decrease is consistent with previous observations for other pore-forming proteins. The fact that not all the protein is internalized, and the remaining *rPfEXP2* is not absorbed by the cells in the following hour (please see **Figure R2**) suggests that cells become non responsive to the action of *rPfEXP2*. This reinforces the idea that the timely delivery of *rPfEXP2* is critical for the rescue of the invasion capacity of EXP2 cKO sporozoites.

Figure R2 - rPfEXP2 in supernatant of HepG2 cells. HepG2 cells were treated with 1 nM of *rPfEXP2* for 5min, 30min and 60min. Supernatant of the cells was collected and processed for WB detection of *rPfEXP2*, left panel. We used the same volume of sample between the different experimental conditions. *rPfEXP2* quantity was normalized to input detection, right panel. Representative WB of 3 independent experiments.

-In the Gcamp6f and GEPII experiments shown in Fig 3c and d, the concentration of rEXP2 used is 10 times that of the highest concentration in the invasion rescue assays. Can these effects be detected at the level of EXP2 treatment used in the rescue assays? Does a 100 nM treatment still rescue invasion? I appreciate that effective concentrations may differ between these experiments with very different readouts, but if so this should be discussed.

The reviewer raises an important concern, the concentration of EXP2 is higher in calcium/potassium reporter experiments (100nM) than that used in rescue experiments (10nM and 1nM). We increased the concentration of recombinant EXP2 to visualize the global changes in the fluorescence of the reporter proteins. In the context of invasion, smaller concentrations of rEXP2 should cause local ion changes in the cell that are enough to induce the necessary mechanisms for sporozoite invasion, but not sufficient to induce changes in global ion concentrations.

-Please show P values for all comparisons, even if they fall above a significance threshold. For example, it is important to state the p value comparing the WT and EXP2 cKO at the 60 min time point in Fig 2d to make the argument that the treatment is rescuing. Same point for the aSMase treatment comparison in Fig 3g, etc.

We have changed the figures, to show the p values for all the comparisons that were performed. This way, reading panels 2d and 3g is easier (please see **Figure R3** below and the figures in the manuscript file).

Figure R3 - Panels from Fig. 2d and Fig. 3g showing all the statistical comparisons performed. All panels from all figures were adjusted as shown.

-It is not always clear which EXP2 antibody is being used in which experiment. For example, in the IFAs in Fig 2b or Sup Fig 2c, it is not clear if this is the rabbit or the mouse anti-PfEXP2. Also, no information is provided in the methods about the RON4 antibody. References should be provided for all non-commercial antibodies if they have been previously published. If they have not been published, information about their generation and validation should be provided. Given that both anti-EXP2 antibodies were raised against *P. falciparum* EXP2, it is important to validate that they cross react with PbEXP2. Is the rabbit anti-PfEXP2 used here the same as that used in Matthews et al 2013?

We thank the reviewer for calling the attention for the lack of information relative to the antibodies used in this study.

The α RON4 antibody used was provided by the laboratory of Maryse Lebrun, used in their recent *Nature Communications* paper¹³. We have added this reference to the manuscript as well as references to all non-commercial antibodies.

We have changed the figure legends to address which antibody was used in which experiment. In short, for experiments in Fig. 1b, 1f, S1a-d, 2b (co-staining with TRAP and UIS4) and S2c, the mouse anti-EXP2 was used. For experiments in Fig. 2a and in 2b (co-staining with anti-RON4), the rabbit anti-EXP2 was used.

To validate both antibodies we performed Western Blot (WB) using *Plasmodium berghei* blood stage parasites and blood from uninfected mice. Also, we used blood stages of EXP2-HA (constructed by Matthews and colleagues¹⁴) parasite line (please see **Figure R4**). We can detect EXP2 during the blood stages using either antibody, in a pattern as was described previously by Matthews et al., 2013¹⁴.

Figure R4 - Detection of EXP2 and EXP2-HA by Western Blot. EXP2 from blood stage parasites (BS) from WT and EXP2-HA parasite lines were analyzed by WB using a mouse anti-EXP2 (left blots) or a rabbit anti-EXP2 (right blot) antibody.

-Why are EXP2-HA parasites used for secretion assays while untagged lines are used for other experiments related to EXP2 section during sporozoite invasion (for example, the IFAs in Fig 2b and Sup Fig 2c)? This crucial experiment seems to be the only place this HA tagged line is used. Do the EXP2 antibodies not work in western blots?

Indeed, the EXP2-HA tagged parasite line was only used for the secretion experiments. We performed Western Blot (WB) using *Plasmodium berghei* blood stage parasites and sporozoites from WT and EXP2-HA (constructed by Matthews and colleagues¹⁴) parasite lines and used rabbit anti-EXP2 and anti-HA to detect EXP2 and EXP2-HA (please see **Figure R5**). We can detect EXP2 during the blood stages using either antibody, in a pattern as was described previously by Matthews et al., 2013¹⁴. In the sporozoite stage, we can detect EXP2-HA using an anti-HA antibody, yet, we cannot detect EXP2 by WB when using anti-EXP2. This is, likely, due to the low abundance of this protein during the sporozoite stage, as shown before¹, combined with the higher antibody sensitivity between the HA tag and the anti-HA antibody.

Figure R5 - Detection of EXP2 and EXP2-HA by Western Blot. EXP2 from blood stage parasites (BS) and sporozoites (Spz) from WT and EXP2-HA parasite lines were analyzed by WB using a rabbit anti-EXP2 (left blot) antibody. The membrane was re-probed with an anti-HA antibody (right blots). EXP2-HA signal in the sporozoites was detected when the membrane was overexposed. MWM: molecular weight marker.

-In fig 2e, expression levels of GAP45 and EXP1 are used to argue that the pattern seen for EXP2 is unique (peaking at 60 minutes). It is claimed from the data that GAP45 is progressively downregulated while EXP1 is progressively upregulated. Are the differences between the time points for GAP45 or EXP1 significant?

We have addressed this and corrected it in Figure 2e (please see **Figure R6**). As the reviewer can observe, the increase in EXP2 mRNA expression is statistically significant, while the decrease of GAP45 and increase of EXP1 are not statistically significant.

Figure R6 - Revised version of Fig. 2e, showing statistical comparison between mRNA expression of invasion-related genes. Bars represent mean±sem and Mann-Whitney *U* test was applied as comparison between time points.

Minor comments:

- line 277: The reference given is #21 but should be #26.
- The methods refer to a number of assays (calcium-free invasion experiments, lines 382-384; ATP depletion experiments, lines 507-509) that do not appear to be present in the manuscript.
- lines 760-762: The legend mentions the blue shading corresponding to the UIS4 cKO but not the green shading that corresponds to the TRAP cKO.
- line 855: knock should be knockdown.
- line 934: There is no figure 2g or j. The authors are referring to 2c and f.

These corrections have been made. We thank the reviewer for pointing out these mistakes.

References

1. Matz, J. M. *et al.* The *Plasmodium berghei* translocon of exported proteins reveals spatiotemporal dynamics of tubular extensions. *Sci. Rep.* **5**, 12532 (2015).
2. Kalanon, M. *et al.* The *Plasmodium* translocon of exported proteins component EXP2 is critical for establishing a patent malaria infection in mice. *Cell. Microbiol.* **18**, 399–412 (2016).
3. Gold, D. A. *et al.* The *Toxoplasma* dense granule proteins GRA17 and GRA23 mediate the movement of small molecules between the host and the parasitophorous vacuole. *Cell Host Microbe* **17**, 642–652 (2015).
4. Bullen, H. E. *et al.* Biosynthesis, Localization, and Macromolecular Arrangement of the *Plasmodium falciparum* Translocon of Exported Proteins (PTEX). *J. Biol. Chem.* **287**, 7871–7884 (2012).
5. Kariu, T., Ishino, T., Yano, K., Chinzei, Y. & Yuda, M. CelTOS, a novel malarial protein that mediates transmission to mosquito and vertebrate hosts. *Mol. Microbiol.* **59**, 1369–79 (2006).
6. Low, L. M. *et al.* Deletion of *Plasmodium falciparum* Protein RON3 Affects the Functional Translocation of Exported Proteins and Glucose Uptake. *MBio* **10**, 1–11 (2019).
7. Risco-Castillo, V. *et al.* Malaria sporozoites traverse host cells within transient vacuoles. *Cell Host Microbe* **18**, 593–603 (2015).
8. Yamauchi, L. M., Coppi, A., Snounou, G. & Sinnis, P. *Plasmodium* sporozoites trickle out of the injection site. *Cell. Microbiol.* **9**, 1215–22 (2007).
9. Douglas, R. G., Amino, R., Sinnis, P. & Frischknecht, F. Active migration and passive transport of malaria parasites. *Trends Parasitol.* **31**, 357–362 (2015).
10. Bando, H. *et al.* CXCR4 regulates *Plasmodium* development in mouse and human hepatocytes. *J. Exp. Med.* **216**, 1733–1748 (2019).
11. Collins, C. R. *et al.* Robust inducible Cre recombinase activity in the human malaria parasite *Plasmodium falciparum* enables efficient gene deletion within a single asexual erythrocytic growth cycle. *Mol. Microbiol.* **88**, 687–701 (2013).
12. Castro-Gomes, T., Corrotte, M., Tam, C. & Andrews, N. W. Plasma membrane repair is regulated extracellularly by proteases released from lysosomes. *PLoS One* **11**, 1–26 (2016).
13. Suarez, C. *et al.* A lipid-binding protein mediates rhoptry discharge and invasion in *Plasmodium falciparum* and *Toxoplasma gondii* parasites. *Nat. Commun.* **10**, 4041 (2019).
14. Matthews, K. *et al.* The *Plasmodium* translocon of exported proteins (PTEX) component thioredoxin-2 is important for maintaining normal blood-stage growth. *Mol. Microbiol.* **89**, 1167–1186 (2013).

Reviewer #1 (Remarks to the Author):

Having reviewed the responses to my own comments and those of the other reviewers I am satisfied with the clarifications made in the revised manuscript and believe that my concerns have been very well addressed. I see no further barriers to the paper advancing to publication and look forward to seeing this really important, and very interesting paper in press.
Jake Baum

Reviewer #3 (Remarks to the Author):

The authors have performed an additional experiment showing that rEXP2 levels decrease to about 50% their initial levels in the culture media of treated HepG2 cells as early as 5 minutes post-addition. I would encourage them to add this new data to the supplementary material and to incorporate their explanation of the data from the response document into the manuscript.

Similarly, I suggest they include their rationalization of why they need to add 10 times as much rEXP2 to produced the effects observed by Gcamp6f and GEPII compared with the invasion rescue assays in the manuscript.

The authors have adequately address my other concerns.

Point-by-point response to the reviewers

We want to thank the reviewers for once again dedicating their time and effort to present their feedback on the manuscript. We have responded to comments of the reviewers and changed the manuscript according to the suggestion present. All the changes made have been highlighted within the manuscript.

Reviewer #1 (Remarks to the Author):

Having reviewed the responses to my own comments and those of the other reviewers I am satisfied with the clarifications made in the revised manuscript and believe that my concerns have been very well addressed. I see no further barriers to the paper advancing to publication and look forward to seeing this really important, and very interesting paper in press.

Jake Baum

We want to thank the reviewer for his positive comments. The suggestions presented by the reviewer for the mechanism are insightful. Their incorporation in the discussion and in the model makes the manuscript a much more thoughtful and overarching text.

Reviewer #3 (Remarks to the Author):

The authors have performed an additional experiment showing that rEXP2 levels decrease to about 50% their initial levels in the culture media of treated HepG2 cells as early as 5 minutes post-addition. I would encourage them to add this new data to the supplementary material and to incorporate their explanation of the data from the response document into the manuscript.

Similarly, I suggest they include their rationalization of why they need to add 10 times as much rEXP2 to produced the effects observed by Gcamp6f and GEPII compared with the invasion rescue assays in the manuscript.

The authors have adequately address my other concerns.

We want to thank the reviewer for the positive remarks. We have added the new data into the Results section of the manuscript and the WBs are shown as the new Supplementary Fig. 2c. Moreover, we have added the rationale of the concentration of protein used in the ion flux experiments to the manuscript. The comments the reviewer made were critical for improving the quality of the text as well as their validity.